# CAMERACTRL: ENABLING CAMERA CONTROL FOR VIDEO DIFFUSION MODELS

**Hao He**[1,2*]   **Yinghao Xu**[3]   **Yuwei Guo**[1,2]
**Gordon Wetzstein**[3]   **Bo Dai**[2]   **Hongsheng Li**[1†]   **Ceyuan Yang**[2†]
[1] The Chinese University of Hong Kong
[2] Shanghai Artificial Intelligence Laboratory
[3] Stanford University

## ABSTRACT

Controllability plays a crucial role in video generation, as it allows users to create and edit content more precisely. Existing models, however, lack control of camera pose that serves as a cinematic language to express deeper narrative nuances. To alleviate this issue, we introduce `CameraCtrl`, enabling accurate camera pose control for video diffusion models. Our approach explores effective camera trajectory parameterization along with a plug-and-play camera pose control module that is trained on top of a video diffusion model, leaving other modules of the base model untouched. Moreover, a comprehensive study on the effect of various training datasets is conducted, suggesting that videos with diverse camera distributions and similar appearance to the base model indeed enhance controllability and generalization. Experimental results demonstrate the effectiveness of `CameraCtrl` in achieving precise camera control with different video generation models, marking a step forward in the pursuit of dynamic and customized video storytelling from textual and camera pose inputs. The project website is at: https://hehao13.github.io/projects-CameraCtrl/.

## 1 INTRODUCTION

Recently, diffusion models have significantly advanced the ability to generate video from text or other input (Blattmann et al., 2023b; Xing et al., 2023; Wu et al., 2023; Ho et al., 2022a; Guo et al., 2023b), with a transformative impact on digital content design workflows. In the applications of practical video generation applications, controllability plays a crucial role, allowing for better customization according to user needs. This improves the quality, realism, and usability of the generated videos. Although text and image inputs are commonly used to achieve controllability, they may lack precise control over visual content and object motion. To address this, some approaches have been proposed, leveraging control signals such as optical flow (Yin et al., 2023; Chen et al., 2023b; Shi et al., 2024), pose skeleton (Ma et al., 2023; Ruiz et al., 2023), and other multi-modal signals (Wang et al., 2024; Ruan et al., 2023), enabling more accurate control for guiding video generation.

However, existing models lack precise control over adjusting or simulating camera viewpoints in video generation. The ability to control the camera viewpoints during the video generation process is crucial in many applications, such as visual reality, augmented reality, and game development. Moreover, skillful management of camera movements enables creators to emphasize emotions, highlight character relationships, and guide the audience's focus, which holds significant value in the film and advertising industries. Recent efforts have been made to introduce camera control in video generation. For example, AnimateDiff (Guo et al., 2023b) incorporates a MotionLoRA module on top of its motion module, enabling some specific types of camera movement. Nevertheless, it struggles to generalize to the camera trajectories customized by users. MotionCtrl (Wang et al., 2023) offers more flexible camera control by conditioning video diffusion models on a sequence of camera pose parameters, but it relies solely on the numerical values of camera parameters without geometric cues

---

*This work was done while the author was an intern at Shanghai Artificial Intelligence Laboratory.
†Corresponding author

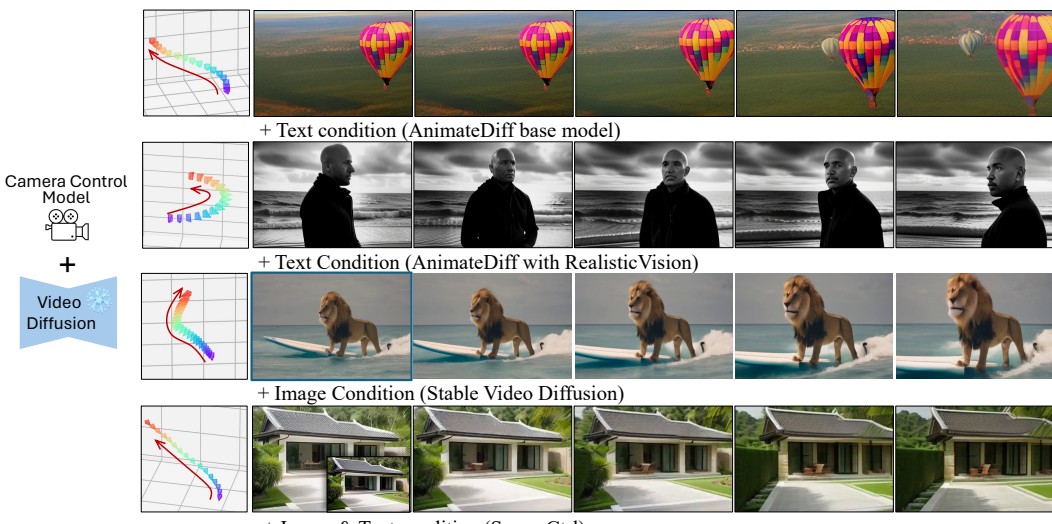

Figure 1: **Illustration of `CameraCtrl`.** It can control the camera trajectory for both general T2V (Guo et al., 2023b) and personalized T2V generation (civitai), shown in the first two rows. Besides, illustrated in the third row, it can be used with I2V diffusion models, like Stable Video Diffusion (Blattmann et al., 2023a). The condition image is the first image of row 3. `CameraCtrl` can also collaborate with other visual controllers, such as the RGB encoder from SparseCtrl (Guo et al., 2023a) to generate videos condition on image and text conditions and manage camera movements.

of camera poses, which is insufficient in ensuring precise camera control. Additionally, MotionCtrl lacks the ability to generalize camera control across other personalized video generation models.

We thus introduce `CameraCtrl`, learning a precise plug-and-play camera pose control module that could control camera viewpoints in video generation. Considering that seamlessly integrating a camera control module into existing video diffusion models is challenging, we investigate how to represent and inject the camera pose effectively. Concretely, we adopt Plücker embeddings (Sitzmann et al., 2021) as the primary form of camera pose conditioning. This choice is attributed to their encoding of geometric interpretations for each pixel in a video frame, offering a comprehensive description of camera pose information. To ensure the applicability and generalizability of our `CameraCtrl` after training, we introduce a camera control module that only takes the Plücker embedding as input and is thus agnostic to the appearance of the training dataset. To evaluate effective training strategies of the camera control model, a comprehensive study is conducted to investigate the effects of various types of training data, ranging from photorealistic to synthetic data. Experimental results suggest that data (*e.g.,* RealEstate10K (Zhou et al., 2018)) with similar appearance to the original base model and diverse camera pose distribution achieve the best trade-off between generalizability and controllability.

We first implement `CameraCtrl` on top of AnimateDiff, enabling precise camera control in text-to-video (T2V) generation across various personalized T2V models (see Fig. 1, rows 1–2). We also integrate `CameraCtrl` with Stable Video Diffusion (Blattmann et al., 2023a) to achieve camera control in the image-to-video (I2V) setting, illustrated in the third row of Fig. 1. In addition, as shown in the last row of Fig. 1, `CameraCtrl` is also compatible with other plug-and-play modules, *e.g.,* SparseCtrl (Guo et al., 2023a) to control video viewpoints under the conditions of both text and structure information, *e.g.,* image.

In summary, our main contributions have three parts:

- We introduce `CameraCtrl`, which empowers video diffusion models with flexible and precise controllability over camera viewpoints.
- The plug-and-play camera control module can be adapted to various video generation models, producing visually appealing camera control.
- We provide a comprehensive analysis of datasets to train the camera control module. We hope this will be helpful for future research in this direction.

## 2 RELATED WORK

**Video generation.** Benefiting from the stability in training and well-established open-sourced communities, recent attempts at video generation, mainly leverage diffusion models (Ho et al., 2020; Song et al., 2020; Peebles & Xie, 2023b). Many recently video diffusion models are the T2V models (Karras et al., 2023; Ruan et al., 2023; Zhang et al., 2023b; He et al., 2022; Chen et al., 2023a; Hong et al., 2022; Yang et al., 2024b). Some approaches seek to replace the guidance signal from text to image, focusing on the I2V setting (Chen et al., 2023b;d; Esser et al., 2023). As a pioneering approach, Video Diffusion Model (Ho et al., 2022b) expands a 2D image diffusion architecture to accommodate video data and jointly train the model on image and video from scratch. To utilize powerful pre-trained image generators, such as Stable Diffusion (Rombach et al., 2022), later works inflate the 2D architecture by interleaving temporal layers between the pre-trained 2D layers and finetune the new model on large video dataset (Bain et al., 2021). Among them, Align-Your-Latents (Blattmann et al., 2023b) efficiently turns a text-to-image (T2I) model into a video generator by aligning independently sampled noise maps. Stable video diffusion (SVD) (Blattmann et al., 2023a) extends Align-Your-Latents by more elaborate training steps and data curation. AnimateDiff (Guo et al., 2023b) utilizes a pluggable motion module to enable high-quality animation creation on personalized image backbones (Ruiz et al., 2023). To enhance temporal coherency, Lumiere (Bar-Tal et al., 2024) replaces the commonly used temporal super-resolution module and directly generates full-frame-rate videos. Other significant attempts include leveraging scalable transformer backbone (Ma et al., 2024), operating in space-temporal compressed latent space, *e.g.*, W.A.L.T. (Gupta et al., 2023) and Sora (Brooks et al., 2024), and using discrete token with language model for video generation (Kondratyuk et al., 2023). Please see (Po et al., 2023) for a comprehensive survey.

**Controllable video generation.** The ambiguity of text or image conditioning alone often leads to weak control for video diffusion models. To provide enhanced guidance, some works adopt other signals, *e.g.*, depth/skeleton sequence, to precisely control the scene/human motion in the generated videos (Guo et al., 2023a; Chen et al., 2023c; Zhang et al., 2023c; Khachatryan et al., 2023; Hu et al., 2023; Xu et al., 2023). Another method (Guo et al., 2023a) utilizes sketch images as the control signal, contributing to high video quality or accurate temporal relationship modeling. In contrast, our work focuses on camera control during the video generation process. AnimateDiff adopts efficient LoRA (Hu et al., 2021) finetuning to obtain model weights specified for different camera movement types. Direct-a-Video (Yang et al., 2024a) proposes a camera embedder to control the camera pose of generated videos, but only conditions three camera parameters, limiting its camera control ability to the most basic types, such as pan left. MotionCtrl (Wang et al., 2023) takes more camera parameters as input to control the camera viewpoints. However, solely relying on the numerical values of camera parameters restricts the accuracy of camera control, and the necessity to fine-tune part of the video diffusion model's parameter can hamper its generalization ability across different video domains. In this study, we aim to control the camera poses during the video generation process precisely, and expect the corresponding model can be used in various video generation models.

## 3 CAMERACTRL

Introducing precise control of the camera into existing video generation methods is challenging, but holds significant value in terms of achieving desired results. To accomplish this, we address this problem by considering three key questions: **(1)** How can we effectively represent the camera condition to reflect the geometric movement in 3D space? **(2)** How can we seamlessly inject the camera condition into existing video generators without compromising frame quality and temporal consistency? **(3)** What type of training data should be utilized to ensure proper model training?

This section is thus organized as follows: Sec. 3.1 presents a brief background discussion of video generation models; Sec. 3.2 introduces the camera representation used by `CameraCtrl`; Sec. 3.3 presents the camera model $\Phi_c$ for injecting camera representation into the video diffusion models. The data selection process is discussed in Sec. 3.4.

### 3.1 PRELIMINARIES OF VIDEO GENERATION

**Video diffusion models.** T2V diffusion models have seen significant advancements in recent years. Some approaches (Singer et al., 2022; Ho et al., 2022b) train video generators from scratch, while

others (Guo et al., 2023b; Blattmann et al., 2023b) utilize powerful T2I diffusion models as the pretrained model and train some temporal blocks on them. Besides, some methods jointly train the video generators using images and videos (Yang et al., 2024b). Despite with different training recipes, these models often adhere to the original formulation used for image generation. Concretely, a sequence of $N$ images (or their latent features) $z_0^{1:N}$ are first added noises $\epsilon$ gradually to a normal distribution at $T$ steps. Given the noised input $z_t^{1:N}$ at the $t$ step, a neural network $\hat{\epsilon}_\theta$ is trained to predict the added noises. During the training, the network tries to minimize the mean squared error (MSE) between its prediction and the ground truth noise scale; the training objective function is formulated as follows:

$$\mathcal{L}(\theta) = \mathbb{E}_{z_0^{1:N}, \epsilon, c_t, t}[\|\epsilon - \hat{\epsilon}_\theta(z_t^{1:N}, c_t, t)\|_2^2], \tag{1}$$

where $c_t$ represents embeddings of the corresponding condition signal, like text prompts.

**Controllable video generation.** In addition to text conditioning, there have been further advancements in enhancing controllability. By incorporating additional structural control signals $s_t$ (*e.g.*, depth maps and canny maps) into the process, controllability for both image and video generation can be enhanced. Typically, these control signals are first fed into an additional encoder $\Phi_s$ and then injected into the generator through various operations (Zhang et al., 2023a; Mou et al., 2023; Ye et al., 2023). Consequently, the objective of training this encoder can be defined as follows:

$$\mathcal{L}(\theta) = \mathbb{E}_{z_0^{1:N}, \epsilon, c_t, s_t, t}[\|\epsilon - \hat{\epsilon}_\theta(z_t^{1:N}, c_t, \Phi_s(s_t), t)\|_2^2]. \tag{2}$$

In this work, we take the camera poses as an additional control signal to the video diffusion models and strictly follow the objective of Eq. (2) to train our camera encoder $\Phi_c$.

## 3.2 CAMERA POSE REPRESENTATION

Before diving into the architecture and training of the camera control module, we first investigate which kind of camera representation could precisely reflect the camera movement in 3D space.

**Camera representation.** Typically, the camera pose refers to the intrinsic and extrinsic parameters, denoted as $\mathbf{K} \in \mathbb{R}^{3 \times 3}$ and $\mathbf{E} = [\mathbf{R}; \mathbf{t}] \in \mathbb{R}^{3 \times 4}$, respectively, where $\mathbf{R} \in \mathbb{R}^{3 \times 3}$ representations the rotation part of the extrinsic parameters, and $\mathbf{t} \in \mathbb{R}^{3 \times 1}$ is the translation part.

To condition a video generator on camera pose, one straightforward choice is to feed raw values of the camera parameters into the generator. However, such a choice may not contribute to accurate camera control for several reasons: **(1)** While the rotation matrix $\mathbf{R}$ is constrained by orthogonality, the translation vector $\mathbf{t}$ is typically unconstrained in magnitude, leading to a mismatch in the learning process of camera control model. **(2)** Direct use of raw camera parameters makes it difficult for the model to correlate these values with image pixels, limiting precise control over visual details. We thus choose Plücker embeddings (Sitzmann et al., 2021) as the camera pose representation. Specifically, for each pixel $(u, v)$ in the image coordinate space, its Plücker embedding is $\mathbf{p}_{u,v} = (\mathbf{o} \times \mathbf{d}_{u,v}, \mathbf{d}_{u,v}) \in \mathbb{R}^6$, where $\mathbf{o} \in \mathbb{R}^3$ is the camera center in world coordinate space, and $\mathbf{d}_{u,v} \in \mathbb{R}^3$ is the direction vector in world coordinate space pointed from the camera center to the pixel $(u, v)$, it is calculated as,

$$\mathbf{d}_{u,v} = \mathbf{R}\mathbf{K}^{-1}[u, v, 1]^T + \mathbf{t}. \tag{3}$$

Then, it is normalized to ensure it has a unit length. For the $i$-th frame in a video sequence, its Plücker embedding can be expressed as $\mathbf{P}_i \in \mathbb{R}^{6 \times h \times w}$, where $h$ and $w$ are the height and width for the frame.

Note that Eq. (3) represents the inverse process of camera projection, which maps a point from the 3D world coordinate space into the pixel coordinate system through the use of matrices $\mathbf{E}$ and $\mathbf{K}$. Thus, compared with the numerical values of extrinsic and intrinsic matrices, the Plücker embedding has more geometric interpretation for each pixel of a video frame then can provide a more informative description of camera pose information to the base video generators. Consequently, it can better adopt the temporal consistency ability of base video generators to generate video clips with a specific camera trajectory. Besides, the value ranges of each item in the Plücker embedding are more uniform, which is beneficial for the learning process of a data-driven model. The illustration of different camera representations are in the Fig. 6, both the camera matrices and the Euler angles are numerical values, while the Plücker embedding is a pixel-wise spatial embedding. After obtaining the Plücker embedding $\mathbf{P}_i$ for the camera pose of the $i$-th frame, we represent the entire camera trajectory of a video as a Plücker embedding sequence $\mathbf{P} \in \mathbb{R}^{n \times 6 \times h \times w}$, where $n$ denotes the total number of frames in a video clip.

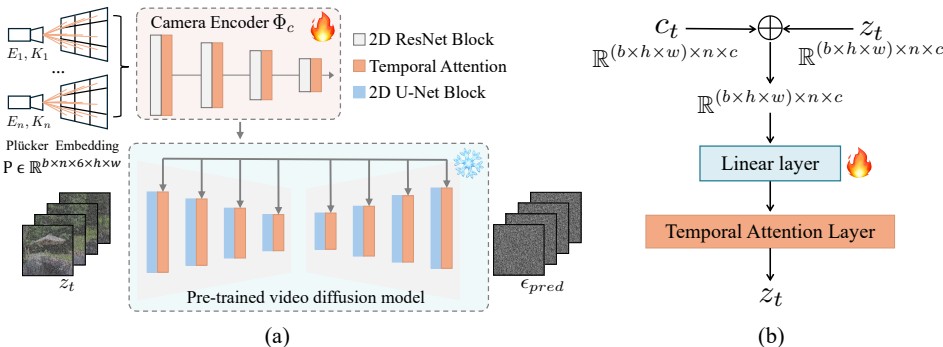

Figure 2: **Framework of `CameraCtrl`.** (a) Given a pre-trained video diffusion model (*e.g.* AnimateDiff (Guo et al., 2023b)) and SVD (Blattmann et al., 2023a), `CameraCtrl` trains a camera encoder on it, which takes the Plücker embedding as input and outputs multi-scale camera representations. These features are then integrated into the temporal attention layers of the U-Net at their respective scales to control the video generation process. (b) Details of the camera injection process. The camera features $c_t$ and the latent features $z_t$ are first combined through the element-wise addition. A learnable linear layer is adopted to further fuse two representations which are then fed into the first temporal attention layer of each temporal block.

### 3.3 CAMERA CONTROLLABILITY INTO VIDEO GENERATORS

Due to the fact that a camera trajectory is parameterized by a Plücker embedding sequence as a pixel-wise spatial ray map, we follow the literature (Zhang et al., 2023a; Mou et al., 2023) by first using an encoder model to extract the features of a Plücker embedding sequence and then fusing the camera features into video generators.

**Camera encoder.** Given a specific camera encoder, we can take both the Plücker embedding sequence and the corresponding image features as its input, like the ControlNet (Zhang et al., 2023a). Alternatively, we can input only the camera features into the camera encoder, as done by the T2I-Adaptor (Mou et al., 2023). Through empirical analysis, we observe that the first approach tends to leak appearance information from the training dataset due to the use of the input image's latent representation. This causes the model to rely on the appearance bias inherent in the training data, thereby limiting its ability to generalize camera pose control across various domains. Therefore, as illustrated in Fig. 2(a), our camera encoder $\Phi_c$ only take the Plücker embedding as input and delivers multi-scale features. Based on the encoder used in T2I-Adaptor, we introduce a camera encoder specifically designed for videos. This camera encoder includes a temporal attention module after each convolutional block, allowing it to capture the temporal relationships between camera poses throughout the video clip. The detailed architecture of camera encoder is in Appendix D.1.

**Camera fusion.** After obtaining the multi-scale camera features, we aim to integrate these features seamlessly into the U-Net architecture of the video diffusion model. Thus, we further investigate the different types of layers in the U-Net, aiming to determine which layer should be used to incorporate the camera information. Recall that the U-Net model contains both spatial and temporal attention. We inject the camera features into the temporal attention blocks. This decision stems from the capability of the temporal attention layer to capture temporal relationships, aligning with the inherent sequential and causal nature of a camera trajectory, while the spatial attention layers always picture the individual frames. This camera feature fusion process is shown in Fig. 2(b). The image latent features $z_t$ and the camera pose features $c_t$ are directly combined through pixel-wise addition. Then, the integrated feature is passed through a linear layer, whose output is fed directly into the fixed first temporal attention layer of each temporal attention module.

### 3.4 LEARNING CAMERA DISTRIBUTION IN DATA-DRIVEN MANNER

Training the aforementioned camera encoder and the fusion linear layers on a video generator usually requires a lot of videos with camera pose annotations. One can obtain the camera trajectory through structure-from-motion (SfM), *e.g.,* COLMAP (Schönberger & Frahm, 2016) for realistic videos while

others could collect videos with ground-truth camera poses from rendering engines, such as Blender. We thus investigate the effect of various training data on the camera-controlled generator.

**Dataset selection.** We aim to select a dataset with appearances that closely match the training data of the base video diffusion models and have as wide a camera pose distribution as possible. We choose three datasets as the candidates: Objaverse (Deitke et al., 2023), MVImageNet (Yu et al., 2023), and RealEstate10K (Zhou et al., 2018). Samples from the three datasets can be found in the Fig. 5.

Indeed, datasets with computer-generated imagery, such as Objaverse (Deitke et al., 2023), exhibit diverse camera distributions since we can control the camera parameters during the rendering process. Yet, these datasets often suffer from a distribution gap in appearance when compared to real-world datasets, such as WebVid10M (Bain et al., 2021), which is used to train the base video diffusion model. When dealing with real-world datasets, such as MVImageNet and RealEstate10K, the distribution of camera parameters is often not very broad. In this case, a balance needs to be found between the complexity of individual camera trajectory and the diversity among multiple camera trajectories. The former ensures that the model learns to control complex trajectories during each training process, while the latter guarantees that the model does not overfit to certain fixed patterns. In reality, while the complexity of camera trajectories in MVImageNet may slightly exceed than that of RealEstate10K for individual trajectories, the trajectories of MVImageNet are typically limited to horizontal rotations. In contrast, RealEstate10K showcases a wide variety of camera trajectories. Considering our goal is to apply the model to a wide range of custom trajectories, we ultimately selected RealEstate10K as our training dataset. Besides, there are some other datasets with characteristics similar to RealEstate10K, such as ACID (Liu et al., 2021) and MannequinChallenge (Li et al., 2019), but their data volume is much smaller than RealEstate10K. We tried to combine them with RealEstate10K to jointly train the `CameraCtrl` model, but found no benefit.

**Measuring the camera controllability.** To monitor the training process of our camera encoder, we design two metrics to measure the camera control quality by quantifying the error between the input camera conditions and the camera trajectory of generated videos. Concretely, we utilize COLMAP (Schönberger & Frahm, 2016) to extract the camera pose sequence of generated videos, which consists of the rotation matrices $\mathbf{R}_{gen} \in \mathbb{R}^{n \times 3 \times 3}$ and translation vectors $\mathbf{T}_{gen} \in \mathbb{R}^{n \times 3 \times 1}$ of a camera trajectory. Furthermore, since the rotation angles and the translation scales are two different mathematical quantities, we measure the angle error and translation error separately and term them as `RotErr` and `TransErr`. Motivated from (Belousov), the `RotErr` is computed by comparing the ground truth rotation matrix $\mathbf{R}_{gt}$ and $\mathbf{R}_{gen}$, formulated as,

$$\texttt{RotErr} = \sum_{i=1}^{n} \arccos \frac{tr(\mathbf{R}_{gen}^{i} \mathbf{R}_{gt}^{i\text{T}})) - 1}{2}, \tag{4}$$

where $\mathbf{R}_{gt}^{i}$ and $\mathbf{R}_{gen}^{i}$ represent the ground truth rotation matrix and generated rotation matrix for the $i$-th frame, respectively. And $tr$ is the trace of a matrix. To quantify the translation error, we use the $L2$ distances between the ground truth translation vector $\mathbf{T}_{gt}$ and $\mathbf{T}_{gen}$, that is,

$$\texttt{TransErr} = \sum_{j=1}^{n} \|\mathbf{T}_{gt}^{i} - \mathbf{T}_{gen}^{i}\|_{2}^{2}, \tag{5}$$

where $\mathbf{T}_{gt}^{i}$ and $\mathbf{T}_{gen}^{i}$ are the ground truth and generated translation vectors in the $i$-th frame. More discussions of `RotErr` and `TransErr` can be founded in Appendix D.5.

## 4 EXPERIMENTS

In this session, we evaluate `CameraCtrl` with other methods and show its applications in different video generation settings. Sec. 4.1 presents the implementation details. Sec. 4.2 compares `CameraCtrl` with baseline methods AnimateDiff (Guo et al., 2023b) and MotionCtrl (Wang et al., 2023). Sec. 4.3 shows the comprehensive ablation studies of `CameraCtrl`. Sec. 4.4 express the various applications of `CameraCtrl`.

### 4.1 IMPLEMENTATION DETAILS

**Base video diffusion model.** In the T2V setting, AnimateDiff V3 (Guo et al., 2023b) is used as the base model. AnimateDiff can be integrated with various T2I LoRAs or base models across

Table 1: **Quantitative comparisons.** MotionCtrl$_{VC}$ and MotionCtrl$_{SVD}$ represent MotionCtrl with VideoCrafter (Chen et al., 2023a) and SVD (Blattmann et al., 2023a) as base model, respectively. Correspondingly, CameraCtrl$_{AD}$ and CameraCtrl$_{SVD}$ denote base models of AnimateDiff and SVD with `CameraCtrl` respectively.

| Method | FVD ↓ | CLIPSIM ↑ | FC ↑ | ODD ↑ | TransErr ↓ | RotErr ↓ | User Preference Rate ↑ (%) |
|---|---|---|---|---|---|---|---|
| AnimateDiff | **1022.4** | 0.298 | 0.930 | **56.4** | Incapable | Incapable | 19.4 |
| MotionCtrl$_{VC}$ | 1123.2 | 0.286 | 0.922 | 42.3 | 14.02 | 1.58 | 37.0 |
| **CameraCtrl$_{AD}$** | 1088.9 | **0.301** | **0.941** | 49.8 | **12.98** | **1.29** | **43.6** |
| SVD | 371.2 | **0.312** | 0.957 | **47.5** | Incapable | Incapable | Incapable |
| MotionCtrl$_{SVD}$ | 386.2 | 0.303 | 0.953 | 41.8 | 10.21 | 1.41 | 26.9 |
| **CameraCtrl$_{SVD}$** | **360.3** | 0.298 | **0.960** | 46.5 | **9.02** | **1.18** | **73.1** |

different genres. This feature helps us evaluate the generalization ability of `CameraCtrl`. When implementing `CameraCtrl` in the I2V setting, SVD (Blattmann et al., 2023a) is the base model.

**Training.** We use the AdamW optimizer to train our model with a constant learning rate of $1 \times 10^{-4}$ (T2V) or $3 \times 10^{-5}$ (I2V). As stated in Sec. 3.4, we choose RealEstate10K as the dataset, with around $65K$ video clips for training. The camera encoder and the linear layers for camera feature injection are trained together with a batch size of 32 for $50K$ steps. More details in Appendix D.2.

**Evaluation metrics.** To ensure that our camera model does not negatively impact the appearance quality of the original video diffusion models, we utilize the Fréchet Video Distance (FVD) (Unterthiner et al., 2018; 2019), CLIPSIM (Radford et al., 2021), and Frame Consistency (FC) (Huang et al., 2023) to evaluate the video appearance quality. Besides, motivated from the Dynamic Degree metric in VBench (Huang et al., 2023), we propose an Object Dynamic Degree (ODD), details in Appendix D.4;, to evaluate the extent of object motion. Additionally, the quality of camera control is evaluated using the metrics RotErr and TransErr introduced in the Sec. 3.4. For the reference videos and (or) the text captions for FVD, CLIPSim, FC, ODD, we random sample 1,000 videos from WebVid10M dataset. For the RotErr, and TransErr, we have randomly chosen 1,000 videos and the corresponding camera poses from the RealEstate10K test set.

## 4.2 COMPARISONS WITH OTHER METHODS

**Quantitative comparison.** To prove the effectiveness of `CameraCtrl`, we compare it with two alternative methods: AnimateDiff with MotionLoRA (Guo et al., 2023b), and MotionCtrl (Wang et al., 2023). Notably, AnimateDiff supports only eight basic camera movements and we do not have the ground truth camera trajectories of these camera movements. Thus, we cannot calculate the RotErr and TransErr for AnimateDiff. Instead, we conducted a user study (details in Appendix E) to assess user preference for camera control capabilities between different models. Besides, we compare `CameraCtrl` with MotionCtrl in both T2V and I2V settings. The quantitative results are shown in Tab. 1. The results of middle block are in the T2V setting, while the results of I2V setting are shown in the bottom block. Compared to AnimateDiff with Motion LoRA and MotionCtrl, it is evident that our approach outperforms them in terms of camera control accuracies (TransErr, RotErr, and user preference). The lower bounds of TransErr and RotErr are listed in the Appendix F.3. Besides, compared with the base models (AnimateDiff and SVD), `CameraCtrl` does not sacrifice the visual quality and the dynamic degree of the gerneated videos, demonstrated by the better or comparable metrics of FVD, CLIPSIM, FC, and ODD.

**Qualitative comparison.** We also provide qualitative comparisons between `CameraCtrl` and MotionCtrl in both T2V and I2V settings in Fig. 3. From the comparisons between the first two rows, we find that MotionCtrl cannot follow the camera condition, it shows the scene rotation, not the camera movement. In contrast, `CameraCtrl` can distinguish the camera movement from the scene motion, strictly following the camera trajectory condition. Besides, MotionCtrl is insensitive to small camera movements. As illustrated in the third row, the result of MotionCtrl shows only forward camera movement, ignoring small movement to the left in the condition trajectory. By comparison, the `CameraCtrl` result in the last row accurately follows both the forward and left camera movements. More qualitative comparison results can be found in Appendix G.

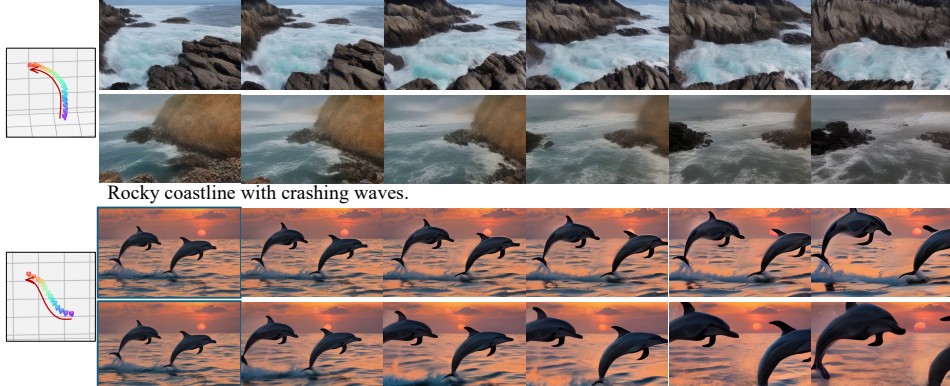

Rocky coastline with crashing waves.

Figure 3: **Qualitative comparisons between CameraCtrl and MotionCtrl.** The first two rows are in the T2V setting, representing MotionCtrl with VideoCrafter and `CameraCtrl` with AnimateDiffV3 as base model, respectively. The last two rows are MotionCtrl and `CameraCtrl` with SVD as base model taking the image as a condition signal. Condition images are the first images of each row.

## 4.3 ABLATION STUDY

We break down the camera control problem into three challenges, regarding the selection of camera representation in Sec. 3.2, the architecture of camera control model in Sec. 3.3, and the learning process of camera control model in Sec. 3.4. In this session, we comprehensively ablate the design choices to each of them using the FVD, `TransErr`, and `RotErr` as the main metrics. All the `CameraCtrl` models in this section are implemented with the AnimateDiffV3 model. Unless otherwise specified, we use the same 1,000 video clips in RealEstate10K dataset as in Sec. 4.2.

**Plücker embeddings represent a camera precisely.** Plücker embeddings naturally serve as a spatial, pixel-wise map with distinct values across locations. As an alternative, we could directly use the numerical values for intrinsic matrix $\mathbf{K}$ and extrinsic matrix $\mathbf{E}$, or convert the rotation matrix of $\mathbf{E}$ into Euler angles. Then, repeating them spatially to form a pixel-wise map with identical content across locations. Another approach is to combine ray directions (varying across pixels) and a repeated camera origin (constant across spatial positions) into a spatial pixel map. Experiment results are illustrated in Tab. 2a – using the Plücker embeddings as the camera representation yields the best camera control results. This stems from Plücker embedding's ability to provide a geometric interpretation for every pixel. By contrast, relying solely on numerical values might lead to numerical mismatches, adversely affecting the camera model's learning efficiency. For the representation using ray direction and camera origin, though it can provide the accurate camera origin information, the repeated the camera origin parameters adds redundancy, potentially misaligning features and hindering the model's understanding of camera motion.

**Noised latents as input limit the generalization.** In ablating the architecture of camera encoders, we differentiate between ControlNet (Zhang et al., 2023a), whose input is the summation of image features and Plücker embedding sqeuence, and the T2I-Adaptor, solely taking Plücker embedding sequence as input. This distinction is crucial as the use of noised latent, mentioned in SparseCtrl (Guo et al., 2023a), has been associated with appearance leakage, effectively limiting the generalization capability of the camera control quality between different domains. Besides, to enhance inter-frame camera coherence, we also consider adding a temporal attention block to each encoder. Thus, when choosing the camera architecture of camera encoder, our experiment covers four configurations: ControlNet, T2I-Adaptor, and their temporal attention-enhanced variants. The ablation result is in the Tab. 2b. With ControlNet as the camera encoder, the appearance quality is suboptimal as shown by the FVD metric in the first two rows. For the models utilizing the T2I-Adaptor, it is observable that the model with additional temporal attention module exhibits better camera control ability. Therefore, we chose the T2I-Adaptor encoder with a temporal attention module as our camera encoder.

**Injecting camera condition into temporal attention.** Next, we investigate where the extracted camera features should be inserted within the pre-trained U-Net architecture. For this purpose, we conduct four experiments to insert the features into the spatial self attention, spatial cross attention, both spatial self and cross attention, and temporal attention layers of the U-Net, respectively. The

Table 2: **Ablation study** on camera representation, condition injection and effect of various datasets.

| Representation type | FVD↓ | TransErr↓ | RotErr↓ | Encoder architecture type | FVD↓ | TransErr↓ | RotErr↓ |
|---|---|---|---|---|---|---|---|
| Raw Values | 230.1 | 13.88 | 1.51 | ControlNet | 295.8 | 13.51 | 1.42 |
| Euler angles | **221.2** | 13.71 | 1.43 | ControlNet + Temporal | 283.4 | 13.13 | 1.33 |
| Direction + Origin | 232.3 | 13.21 | 1.57 | T2I Adaptor | 223.4 | 13.27 | 1.38 |
| **Plücker embedding** | 222.1 | **12.98** | **1.29** | T2I Adaptor + Temporal | **222.1** | **12.98** | **1.29** |

|  (a) How to represent camera parameters. |  | (b) Camera encoder architecture. |  |

| Attention | FVD↓ | TransErr↓ | RotErr↓ | Datasets | FVD↓ | TransErr↓ | RotErr↓ |
|---|---|---|---|---|---|---|---|
| Spatial Self | 241.2 | 14.72 | 1.42 | Objaverse | 1435.4 | Incapable | Incapable |
| Spatial Cross | 237.5 | 14.31 | 1.51 | MVImageNet | 1143.5 | 13.87 | 1.52 |
| Spatial Self + Cross | 240.1 | 14.52 | 1.60 | RealEstate10K + ACID | 1102.4 | 13.48 | 1.41 |
| **Temporal** | **222.1** | **12.98** | **1.29** | **RealEatate10K** | **1088.9** | **12.99** | **1.39** |

| (c) Where to inject camera representations. | (d) Effect of datasets. |

results are presented in the Tab. 2c, indicating that inserting camera features into the temporal attention layers yields better outcomes. This improvement can be attributed to the fact that camera motion typically induces global view changes across frames. Integrating camera poses with the temporal blocks of the U-Net resonates with this dynamic nature.

**Videos with similar appearance distribution and diverse camera help controllability.** To test our argument on dataset selection, as discussed in Sec. 3.4, we train `CameraCtrl` using different datasets. The Objaverse (Deitke et al., 2023) dataset, has the widest distribution of camera poses but has a significantly different appearance from WebVid10M. For real-world datasets, compared to MVImageNet, RealEstate10K possesses a more diverse range of camera trajectories. We evaluate these models using diverse data sources, including WebVid10M for FVD, and MannequinChallenge (Li et al., 2019) camera trajectories for `TransErr` and `RotErr`. The results, as shown in Tab. 2d, display that compared to RealEstate10K, both FVD scores and camera errors are significantly higher with MVImageNet. For Objaverse, COLMAP struggles to extract a sufficient number of camera poses to yield meaningful `TransErr` and `RotErr` metrics. One possible reason for this result is that the difference in the dataset appearance may prevent the model from effectively distinguishing between camera pose and appearance, making a difficult for COLMAP to estimate the camera pose. We also jointly train `CameraCtrl` with RealEstate10K and a similar dataset ACID, but there is no improvement regarding the camera control ability. This result indicates that to improve `CameraCtrl` further, a dataset with a larger camera distribution is needed.

## 4.4 APPLICATIONS OF CAMERACTRL

**Applying `CameraCtrl` to different video generators.** As detailed in Sec. 3.3, our camera control model exclusively uses Plücker embeddings as input, making it independent of the appearance of the training dataset. Besides, as mentioned in Sec. 3.4, we select a dataset with an appearance closely resembling that of the training data of the base video generators. Benefiting from these design choices, `CameraCtrl` can focus solely on learning camera control-related information. This enables its application across various video generators. We demonstrate this in Fig. 4, Appendix H.1, and Appendix H.2. In the T2V setting, we implement `CameraCtrl` based on AnimateDiff. The result shown in the first row adopts the vanilla AnimateDiff model depicting a natural scene. Rows two and three showcase results generated by AnimateDiff embedded with other personalized image generators (Realistic Vision (civitai) and ToonYou (BradCatt)). Row two represents a video style divergent from the typical reality, the buildings of a cyberpunk city. The third row expresses a video of a cartoon character. Besides, we implement `CameraCtrl` with SVD and sample a video in the I2V setting, shown in the last row. Across these varied video generation types, `CameraCtrl` consistently demonstrates effective control over the camera trajectories, showcasing its broad applicability and effectiveness in enhancing video narratives through dynamic camera trajectory control.

**Integrating `CameraCtrl` with other video control methods.** Thanks to the plug-and-play nature of our method, not only can it be used during the generation processes of different base video generators, but it can also be integrated with other video generation control techniques together

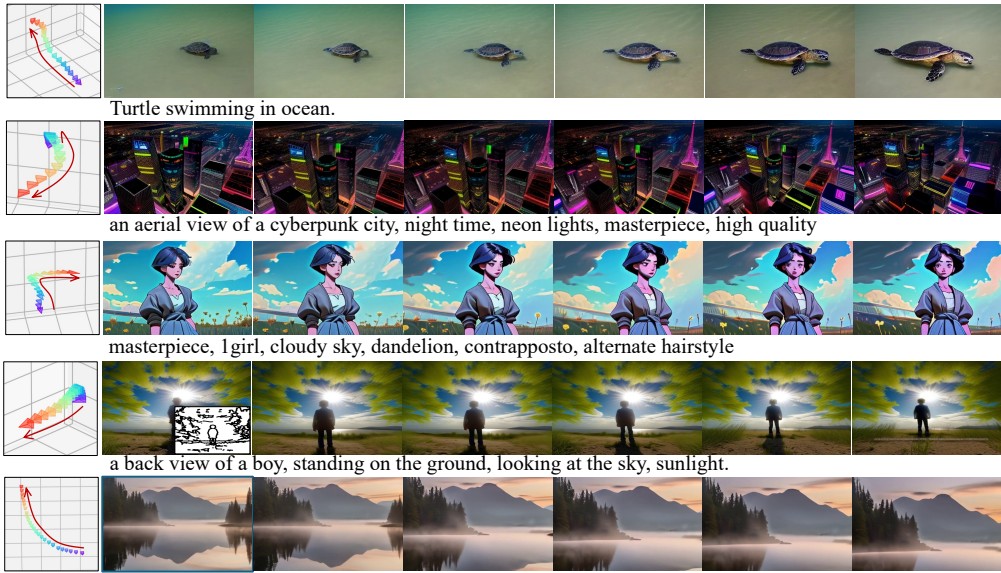

Figure 4: **Applications of `CameraCtrl`.** The first row represents a video generated by the base AnimateDiff. The Following two rows showcase the results of two personalized T2V generators, RealisticVision and ToonYou. The fourth row expresses the video generated by CameraCtrl integrated with another video control method, SparseCtrl (Guo et al., 2023a). The video of the last row is produced by a I2V generator, SVD, taking the first image of last row as a condition.

to produce videos. For example, we utilize SparseCtrl (Guo et al., 2023a), a recent approach that controls the overall video generation by manipulating a few sparse frames. This control can be based on RGB images, sketch maps, or depths. Here, we adopt the RGB encoder and sketch encoder of SparseCtrl, results are shown in the last row of Fig. 1 and the fourth row of Fig. 4, using RGB and sketch encoder of SparseCtrl, respectively. As illustrated in these two videos, this approach has a high level of consistency between the scenes and objects in the generated video and those in the reference frame. Additionally, it is obvious that the camera movements of the generated videos possess a high level of alignment with the provided camera trajectories. The successful integration of CameraCtrl with SparseCtrl further demonstrates the generalization capabilities of CameraCtrl and enhances its application scenarios. More visual results of this part can be found in Appendix H.3.

## 5 DISCUSSION

In this work, we present `CameraCtrl`, a method that addresses the limitations of existing models in precise camera control for video generation. By learning a plug-and-play camera control module, `CameraCtrl` enables accurate control over camera viewpoints. Plücker embeddings are adopted as the primary representation of camera parameters, providing a comprehensive description of camera pose information by encoding geometric interpretations. Through a comprehensive study on training data, it is found that using data with similar appearance to the base model and diverse camera pose distributions, such as RealEstate10K, achieves the best trade-off between generalizability and controllability. Experimental results demonstrate the effectiveness of `CameraCtrl`.

**Ethics Statement.** `CameraCtrl` enhances video generation technology by providing precise control over camera viewpoints, significantly improving the realism and interactivity of many applications. Conversely, `CameraCtrl` could raise some ethical concerns, particularly in the realms of privacy and the potential for creating some misleading contents. There is a critical need for ethical oversight and more advanced deepfake detectors to manage these risks and ensure the proper usage of `CameraCtrl`.

**Reproducibility Statement.** We provide detailed implementation of our training method in the main text Sec. 4.1 and Appendix D.2. We also provide the model architecture of camera encoder in Appendix D.1.

## 6 ACKNOWLEDGEMENT

This project is funded in part by National Key R&D Program of China Project 2022ZD0161100, and by NSFC-RGC Project N_CUHK498/24. Hongsheng Li is a PI of CPII under the InnoHK.

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

## A    APPENDIX / SUPPLEMENTAL MATERIAL

This supplementary material provides discussion of `CameraCtrl` and other methods, more discussions on data selection, implementation details, details of user study, additional ablation experiments, more qualitative comparisons, and more visualization results of `CameraCtrl`.

In all visual results, the first image in each row represents the camera trajectory of a video. Each small tetrahedron on this image represents the position and orientation of a camera for one video frame. Its vertex stands for the camera location, while the base represents the imaging plane of the camera. The red arrows indicate the movement of camera **position** but do **not** depict the camera rotation. The camera rotation can be observed through the orientation of the tetrahedrons. **For a clearer understanding of the camera control effects, we highly recommend that readers watch the videos provided in our supplementary file.**

The organization of this supplementary material is as follows: Appendix B gives some discussions between `CameraCtrl` and concurrent works. Appendix C presents more discussions on the dataset selection process. Appendix D gives more implementation details. The details of the user study are shown in Appendix E. Appendix F depicts extra experiment results on the model architecture, camera representation, and the lower bound of the `RotErr` and `TransErr`. Then, we provide more qualitative comparisons between `CameraCtrl` with AnimateDiff and MotionCtrl in Appendix G. After that, more visualization results are showcased in Appendix H. Finally, we provide some failure cases in Appendix I.

## B    DISCUSSION WITH CONCURRENT CAMERA CONTROL WORKS

Recent works have explored camera control in video generation, addressing different aspects of the challenge. VD3D (Bahmani et al., 2024) integrates camera control into a DiT-based (Peebles & Xie, 2023a; Menapace et al., 2024) model with a novel camera representation module in spatiotemporal transformers. CamCo (Xu et al., 2024b) leverages epipolar constraints for 3D consistency in image-to-video generation. CVD (Kuang et al., 2024) uses the camera control method and extends it to enable multi-view video generation with cross-view consistency. Recaprture (Zhang et al., 2024) enables video-to-video camera control, effectively modifying viewpoints in existing content. However, it's limited to simpler scenes and struggles with complex or dynamic environments. Cavia (Xu et al., 2024a) enhances multi-view generation through training on diverse datasets, improving cross-view consistency. (Cheong et al., 2024) improves camera control accuracy using a classifier free guidance Ho & Salimans (2022) like mechanism in a DiT-based Zheng et al. (2024) model. Despite the numerous works addressing camera control in the video generation process, to our best knowledge, `CameraCtrl` is among the early methods to achieve accurate camera control in video generation models. It provides valuable insights and a solid foundation for future advancements in related fields, such as video generation, as well as 3D and 4D content generation.

## C    MORE DISCUSSIONS ON DATASET SELECTION

When selecting the dataset for training our camera control model, we first choose three datasets as candidates, they are Objaverse (Deitke et al., 2023), MVImageNet (Yu et al., 2023), and RealEstate10K (Zhou et al., 2018).

For the Objaverse dataset, its images are rendered with software like Blender, enabling highly complex camera poses. However, as seen in row one to row three of Fig. 5, its content mainly focuses on objects against white backgrounds. In contrast, the training data for many video diffusion models, such as WebVid-10M (Bain et al., 2021), encompasses both objects and scenes against more intricate backgrounds. This notable difference in appearance can detract from the model's ability to concentrate solely on learning camera control. In our initial trial, We tried to train the `CameraCtrl` with the Objaverse dataset, the resulting model can control the camera trajectory in the Objaverse-like video (single object with white background) well. In other domains, however, the camera control model cannot generalize well in controlling the camera viewpoints during the video generation process.

Table 3: **Output feature shapes of each layer (encoder scale) of camera encoder.** And $c = 6 \times 8 \times 8 = 384$, $c_1, c_2, c_3, c_4$ are equal to the channels numbers of the corresponding U-Net output feature with the same resolution. For examble, with a stable video 1.5 model, $c_1, c_2, c_3, c_4$ equal to 320, 640, 1280, 1280.

| input | $b \times n \times 6 \times h \times w$ |
|---|---|
| Pixel unshuffle | $b \times n \times c \times \frac{h}{8} \times \frac{w}{8}$ |
| $3 \times 3$ conv layer | $b \times n \times c_1 \times \frac{h}{8} \times \frac{w}{8}$ |
| Encoder scale 1 | $b \times n \times c_1 \times \frac{h}{8} \times \frac{w}{8}$ |
| Encoder scale 2 | $b \times n \times c_2 \times \frac{h}{16} \times \frac{w}{16}$ |
| Encoder scale 3 | $b \times n \times c_3 \times \frac{h}{32} \times \frac{w}{32}$ |
| Encoder scale 4 | $b \times n \times c_4 \times \frac{h}{64} \times \frac{w}{64}$ |

For MVImageNet data, it has some backgrounds and complex individual camera trajectories. Nevertheless, as demonstrated in row four to row six of Fig. 5, most of the camera trajectories in the MVImageNet are horizontal rotations. Thus, its camera trajectories lack diversity, which could lead the model to converge to a fixed pattern.

Regarding RealEstate10K data, as shown in row seven to row nine of Fig. 5, it features both indoor and outdoor scenes and objects. Besides, each camera trajectory in RealEstate10K is complex and there exists a considerable variety among different camera trajectories. Therefore, we choose the RealEstate10K dataset to train our camera control model. There are other datasets that possess a similar camera trajectory with the RealEstate10K dataset, like the ACID (Liu et al., 2021) and MannequinChallenge data (Li et al., 2019), but with fewer data samples. We tried to train the `CameraCtrl` using the RealEstate10K and the ACID but did not find an improvement in the camera control accuracy, as shown in Tab. 2d. This result indicates that the current bottleneck of the camera control accuracy may lie in the complexity of the camera pose distribution.

## D  MORE IMPLEMENTATION DETAILS

### D.1  CAMERA ENCODER $\Phi_c$ ARCHITECTURE.

As stated in Sec. 3.3, we take a temporal attention-enhanced T2I Adaptor (Mou et al., 2023) encoder as our camera encoder $\Phi_c$ to extract the camera features from Plücher embeddings. Generally, the camera encoder consists of a pixel unshuffle layer, a convolution layer, and 4 encoder scales. It takes a batch of Plücker embedding sequence $\mathbf{P} \in \mathbb{R}^{b \times n \times 6 \times h \times w}$ where $b, n, h, w$ represent the batch size, number of frames in a video clip, the height and width of the video clip, respectively as input, and output multi-scale camera features. The output feature shapes of each layer are listed in Tab. 3.

Besides, each encoder scale is composed of one downsample ResNet block (Mou et al., 2023) (except for the encoder scale 1) and one ResNet block, each block is followed by one temporal attention block (Guo et al., 2023b). More specifically, the temporal attention block consists of a temporal self-attention layer, layer normalizations and position-wise MLP as follows:

$$\zeta \leftarrow x + \text{PosEmb}(x) \tag{6}$$

$$\zeta_1 \leftarrow \text{LayerNorm}(\zeta) \tag{7}$$

$$\zeta_2 \leftarrow \text{MultiHeadSelfAttention}(\zeta_1) + \zeta \tag{8}$$

$$\zeta_3 \leftarrow \text{LayerNorm}(\zeta_2) \tag{9}$$

$$x \leftarrow \text{MLP}(\zeta_3) + \zeta_2, \tag{10}$$

$$\tag{11}$$

where $\text{PosEmb}$ is the temporal positional embedding.

### D.2  TRAINING.

We use the LAVIS (Li et al., 2023) to generate the text prompts for each video clip of the used dataset (Objaverse, MVImageNet, RealEstate10K, and ACID). For the text-to-video (T2V) setting,

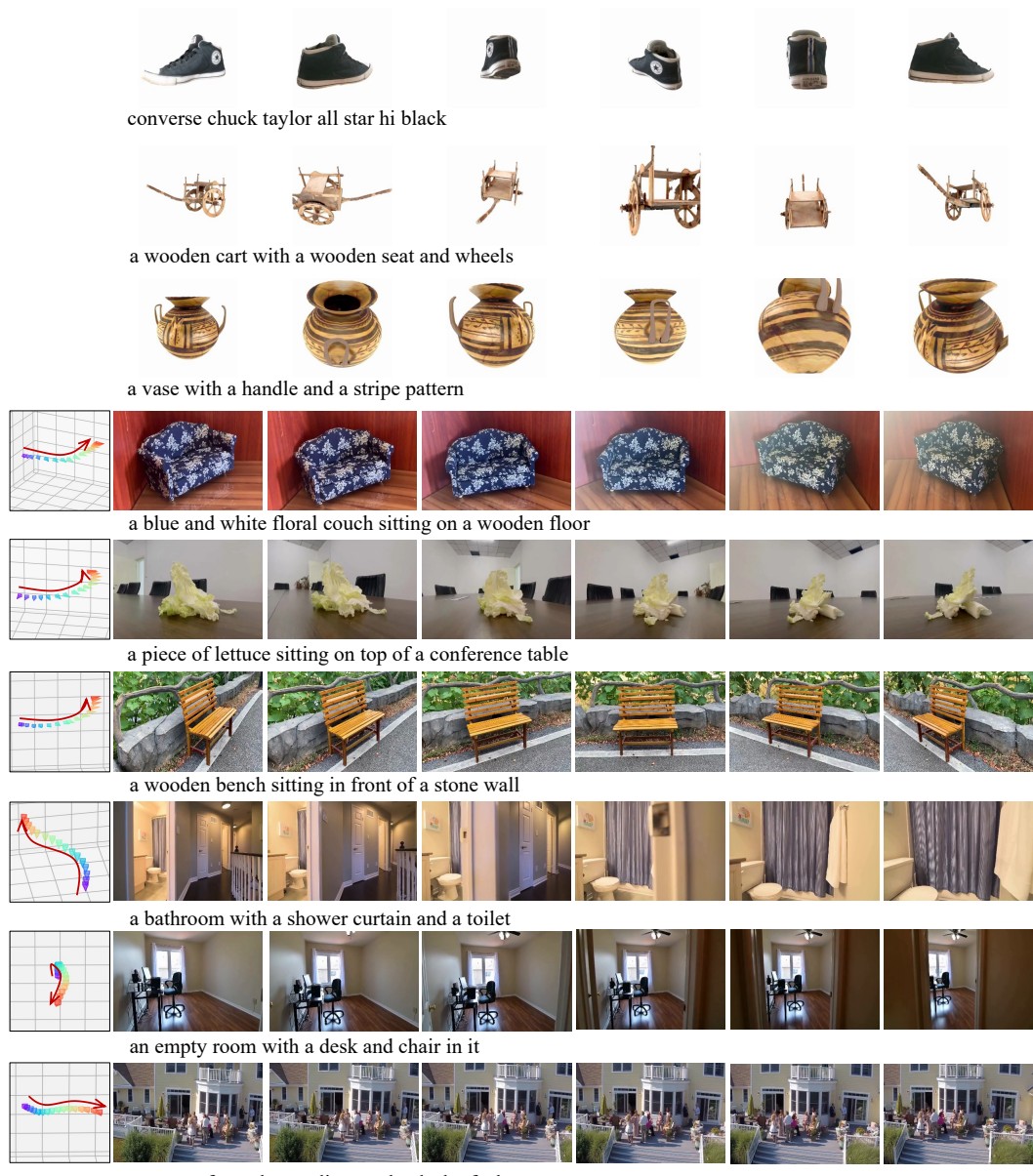

Figure 5: **Samples of different datasets.** Rows 1 to row 3 are samples from the Objaverse dataset, which has random camera poses for each rendered image. Rows 4 to row 6 show the samples from the MVImageNet dataset. Samples of the RealEstate10K dataset are presented from rows 7 to row 9.

we used AnimateDiffV3 as the base video generation model. To let the camera control model better focus on learning camera poses, similar to AnimateDiff (Guo et al., 2023b), we first train an image LoRA on the images of the RealEstate10K dataset. Then, based on the AnimateDiff model enhanced with LoRA, we train the camera control model. Note that, after the camera control model is trained, the image LoRA can be removed. For each training sample of the CameraCtrl, we sample 16 images from one video clip with the sample stride equal to 8, then resize their resolution to $256 \times 384$. For the data augmentation, we use the random horizontal flip for both images and poses with a 50 percent probability. The Adam optimizer is adopted to train the models, with a constant learning rate $1e^{-4}$, $\beta_1$=0.9, $\beta_2$=0.99, weight decay equals 0.01. We use a *linear* beta noise schedule, where $\beta_{start} = 0.00085$, $\beta_{end} = 0.012$, and $T = 1000$. We use 16 80G NVIDIA A100 GPUS to train the CameraCtrl models with a batch size of 2 per GPUS for 50K steps, taking about 25 hours.

For the Image-to-Video (I2V) setting, we use the Stable Video Diffusion (SVD) as the base video generator. We directly train the camera encoder and the merge linear layer on top of the SVD. For each training sample, we sample 14 images for one video clip with the sample stride equal to 8, then resize their resolution to $320 \times 576$. Then Adam optimizer is utilized with a constant learning rate $3e^{-5}$, $\beta_1$=0.9, $\beta_2$=0.99, weight decay equals to 0.01. Following SVD, we used the EDM (Karras et al., 2022) noise scheduler and set all the hyper-parameters equal to SVD. We used 32 80G NVIDIA A100 GPUS to train the models with a batch size of 1 per GPUS for 50K steps, taking about 40 hours.

### D.3 INFERENCE.

By utilizing structure-from-motion methods such as COLMAP (Schönberger & Frahm, 2016), along with existing videos, we can extract the camera trajectory within a video. This extracted camera trajectory can then be fed into our camera control model to generate videos with similar camera movements. Additionally, we can also design custom camera trajectories to produce videos with desired camera movement. During the inference, we use different guidance scales for different domains' videos and adopt a constant denoise step 25 for all the videos.

### D.4 OBJECT DYNAMIC DISTANCE (ODD) METRIC

We first utilize the Grounded-SAM-2 (Ren et al., 2024) to segment the main object in a video. Then, following the dynamic degree in VBench, we use RAFT (Teed & Deng, 2020) to estimate the optical flow, and only keeping the estimated flows belongs to the main object. Then, following the dynamic degree metric, these optical flows are taken as the basis to determination whether the video is static. The final object dynamic degree score is calculated by measuring the proportion of nonstatic videos generated by the model.

### D.5 MORE DETAILS OF `RotErr` AND `TransErr`

When using the COLMAP to extract the camera poses of generated videos, it is not very stable to generate the reliable camera pose sequence. Thus, when COLMAP fails, we have manually filtered these failed video clips and not added them in the calculation of `RotErr` and `TransErr`. Besides, since the COLMAP is scale invariant, there may have some scale issues of the generated camera poses. These scale issues only have some impact on the calculation of `TransErr`, not the `RotErr`. To deal with with this scale issue, we have performed some postprocessings to the COLMAP results. Specifically, we first compute the relative poses of the ground truth and generated camera poses by setting the homogeneous extrinsic matrix of first frame as a $4 \times 4$ identity matrix. Then, normalizing the scale of the COLMAP results with the ground truth camera trajectory. Concretely, we calculate the translation gap between the first two frames for both the generated and ground truth camera poses to obtain a rescale factor. We then normalize other generated camera poses with this rescale factor to align the scales of the two camera trajectories. This normalization helps alleviate the scale problem in COLMAP results, making our evaluation metrics more convincing.

## E DETAILS OF USER STUDY

The camera error `TransErr` and `RotErr` proposed in Sec. 3.4 need COLMAP to extract the camera poses of the generated videos. However, the COLMAP can not extract precise camera poses in short videos (16 frames in T2V, 14 frames in I2V) stably. To compare the camera control quality between `CameraCtrl` with AnimateDiff and MotionCtrl from another perspective, we conduct some user studies. Specifically, since the AnimateDiff is only able to generate videos with eight base camera movements in the T2V setting, we sample these three methods with base camera movements and let the user watch the video to decide which video is more in line with the condition camera trajectory. Then calculating the approving rate for each method, the results are in the User Preference Rate column in Tab. 1's middle block. Besides, we employ some complex camera trajectories extracted from the test set of RealEstate10K to condition the MotionCtrl and `CameraCtrl` in the T2V setting. The generated videos are sent to users to decide which one has a better camera trajectory alignment with the reference videos. The user preference rate for MotionCtrl and `CameraCtrl` are 27.6% and 72.4%, respectively.

Table 4: Ablation study of the camera feature injection place.

| Injection Place | FVD↓ | TransErr↓ | RotErr↓ |
|---|---|---|---|
| U-Net Encoder | **210.9** | 13.91 | 1.51 |
| **U-Net Encoder + Decoder** | 222.1 | **12.98** | **1.29** |

Figure 6: **Different camera representation.** The left subfigure row shows the camera represented using the intrinsic $K_i$ and the extrinsic matrices $E_i$ (composed of rotation matrix $R_i$ and the translation vector $t_i$). The middle subfigure give the camera representation of converting the rotation matrix $R_i$ into Euler angles $\alpha_i, \beta_i, \gamma_i$. Plücker embedding are given in the right subfigure, the intrinsic and extrinsic matrices are converted into the Plücker embeddings to form a pixel-wise spatial embedding. While the left and middle camera representations are not a pixel-wise camera representations naturally.

After that, in the I2V setting, we sample the MotionCtrl and `CameraCtrl` with complex camera trajectories extracted from the RealEstate10K dataset. With these videos, another user study is conducted to let the user choose which video has the better camera trajectory condition performance. Results are shown in the User Preference Rate column of the bottom block of Tab. 1. These user study results further demonstrate the the superiority of `CameraCtrl` in controlling the camera trajectory during the video generation process. We invite 50 users to conduct all the user studies. Considering the difference between the education levels of these users, we design the user studies as easily as possible to get more reliable results.

## F    EXTRA EXPERIMENTS

### F.1    EXTRA ABLATION STUDY

**Injecting camera features into both encoder and decoder of U-Net.** In the vanilla T2I-Adaptor (Mou et al., 2023), the extracted control features are only fed into the encoder of U-Net. In this part, we explore whether injecting the camera features into both the U-Net encoder and decoder could result in performance improvements. The experiment results are shown in Tab. 4. The improvements of `TransErr` and `RotErr` indicate that compared to only sending camera features to the U-Net encoder, injecting the camera features to both the encoder and decoder enhances camera control accuracy. This result could be attributed to the fact that similar to text embedding, Plücker embedding inherently lacks structural information. Such that, this integrating choice allows the U-Net model to leverage camera features more effectively. Therefore, we ultimately choose to feed the camera features to both the encoder and decoder of the U-Net.

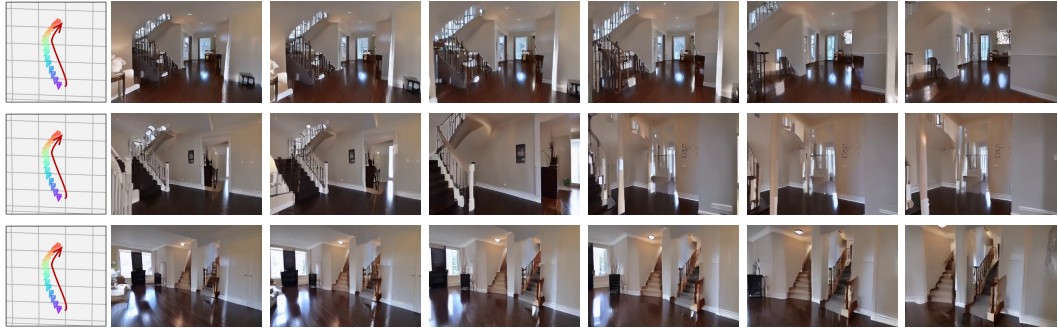

There is a stair to the upper floors and tables and chairs

Figure 7: **Qualitative comparison of using different camera representations.** The first row shows the result using the raw camera matrix values as camera representation. Result of the second row adopts the ray directions and camera origin as camera representation. The last row exhibits the result taking the Plücker embedding as the camera representation. All the results use the same camera trajectory and the text prompt.

Table 5: Lower bound of `TransErr` and `RotErr` on RealEstate10K test set.

|  | TransErr↓ | RotErr↓ |
| --- | --- | --- |
| Lower Bounds | 6.93 | 1.02 |

## F.2 Qualitative comparison on different camera representation

Here, we provide the qualitative comparison using different camera representations, results are shown in Fig. 7. The provided camera trajectory primarily moves forward, with a rightward shift at the end. From the figure, it can be seen that when using the raw camera matrix values as camera representation, the model ignores the final rightward movement. With the hybrid camera pose representation, the model exhibits an abrupt shift in the last few frames to achieve the rightward movement. In contrast, using the Plücker embedding as the camera representation results in a smoother generated video, with the final rightward movement appearing natural and seamless. These results further demonstrate the effectiveness of using Plücker embedding as the camera representation.

## F.3 Lower bound of TransErr and RotErr on RealEstate10K test set

Since the COLMAP is not 100 percent accuracy, we need to know the lower bounds of the `TransErr` and `RotErr` metrics. With the sampled video clips (each has 16 frames) in the RealEstate10K test set, we run the COLMAP on these video clips to get the estimated camera poses. Using these camera poses and the ground truth camera poses, we calculate the `TransErr` and `RotErr`, results are shown in the Tab. 5

## G More Qualitative Comparisons

In this section, we first provide more qualitative comparisons between `CameraCtrl` with Animate-Diff using the base camera trajectories in the test-to-video (T2V) setting. Then, in the T2V setting, we also deliver more qualitative comparisons between `CameraCtrl` and MotionCtrl with the complex camera trajectories extracted from the RealEstate10K test set. Finally, more qualitative comparisons between `CameraCtrl` and MotionCtrl in the image-to-video (I2V) setting are given.

### G.1 Qualitative comparisons in the T2V setting

**Comparisons between AnimateDiff and `CameraCtrl`.** Results are shown in Fig. 8. In row 1, we find that the generated video of AnimateDiff shows the camera movement of pan up not the given camera movement pan down. In contrast, the video generated by `CameraCtrl` in row 2 follows

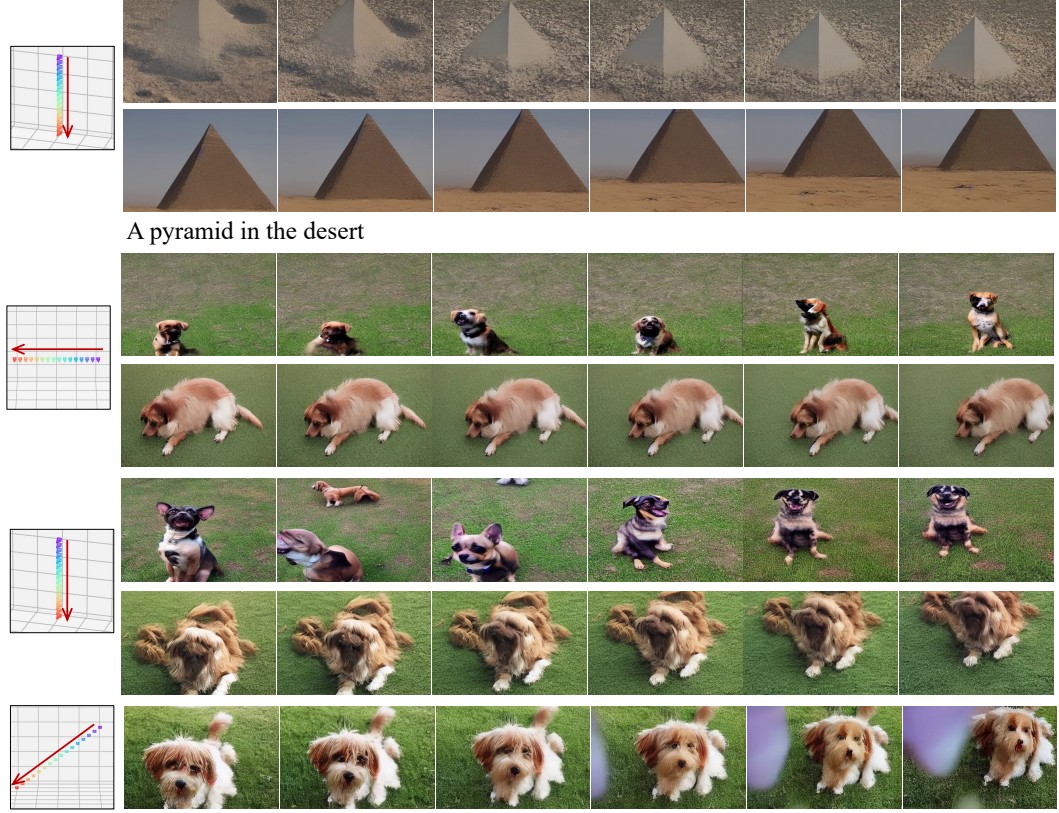

A pyramid in the desert

A cute dog sitting on the green grass.

Figure 8: **Qualitative comparison between AnimateDiff and `CameraCtrl`**.Results of rows 1, 3, and 5 are from AnimateDiff. Results of `CameraCtrl` are shown in rows 2, 4, 6, 7. Rows 1 and 2 use the same camera trajectory, pan down. Camera trajectory pan left is adopted by rows 3 and 4. For rows 5 and 6, the camera trajectory pan down is utilized. The result of the last row is generated with the camera trajectory pan left down. Rows 1 and 2 condition on the same text prompt, while rows 3 to 7 condition on another text prompt.

the desired camera movement. Thus, we can conclude that in some situations, AnimateDiff cannot distinguish the object movement from the camera movement. Results of rows 3 and 5 exhibit that, though sometimes AnimateDiff can generate the videos with the desired camera movement, it cannot keep the object consistent during the whole video. By comparison, `CameraCtrl` can generate videos with consistent contents and strictly follows the condition camera trajectory, illustrated in rows 4 and 6. Besides, AnimateDiff only supports some simple camera trajectories. For other more complex camera trajectories, it does not support even a combination of two base camera trajectories, like the combination of pan left and pan down, while `CameraCtrl` can support this trajectory, shown in the last row of Fig. 8.

**Comparisons between MotionCtrl and `CameraCtrl`.** In Fig. 9, we provide more qualitative comparisons between the MotionCtrl and `CameraCtrl` in the T2V setting. For the trajectories with translation or rotation to a small extent, like the left camera translation in the first trajectory (rows 1 and 2) and the left rotation at the beginning of the second trajectory (rows 3 and 4), MotionCtrl does not very sensitive to them. It only focuses on the main camera movement, the forward translation. By contrast, the generated videos of `CameraCtrl` (rows 2 and 4) accurately obey these small camera trajectories. For the third (rows 5 and 6) and the fourth (rows 7 and 8) camera trajectories, they contain both camera rotation and translation. For the videos generated by MotionCtrl (rows 5 and 7), however, it focuses more on the camera rotation and ignores the camera translation. In contrast, `CameraCtrl` can make a good balance between the camera rotation and translation and generate satisfactory videos, shown in rows 6 and 8 of Fig. 9.

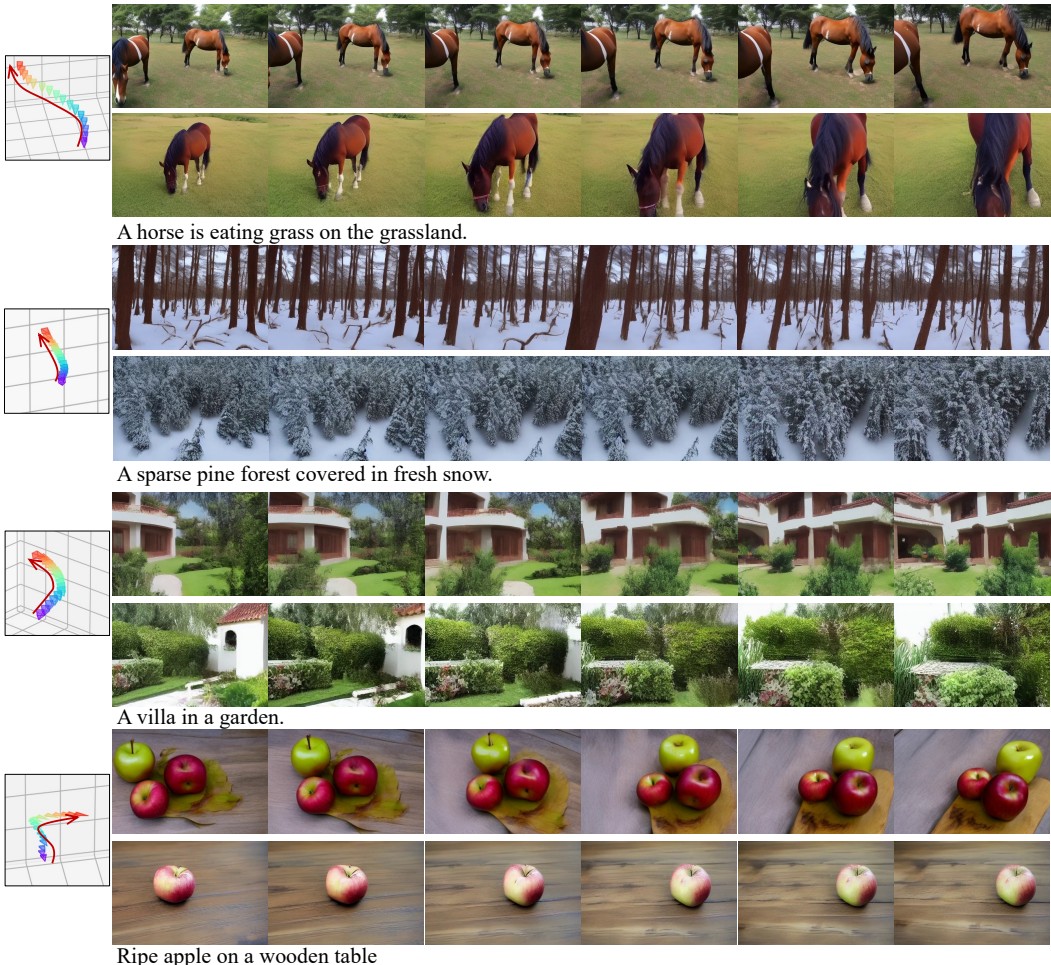

A horse is eating grass on the grassland.

A sparse pine forest covered in fresh snow.

A villa in a garden.

Ripe apple on a wooden table

Figure 9: **Qualitative comparison between MotionCtrl and `CameraCtrl` in T2V setting.** Results of rows 1, 3, 5, and 7 are generated by MotionCtrl, while the results of `CameraCtrl` are shown in rows 2, 4, 6, and 8. Every two adjacent rows use the same text prompt and camera trajectory.

### G.2 QUALITATIVE COMPARISONS IN THE I2V SETTING

Similar to the results of T2V, MotionCtrl still cannot handle the small camera movement well in the I2V setting.

In Fig. 10, for the results of the first two camera trajectories, compared to the results of our `CameraCtrl` (rows 2 and 4), videos generated by MotionCtrl (rows 1 and 3) ignore the small camera rotation at the very beginning of the camera trajectories. For the third and the fourth camera trajectories, the camera movement extent of MotionCtrl results (rows 5 and 7) is rather less than that of the `CameraCtrl` results (rows 6 and 8). The results of `CameraCtrl` can reveal the practical camera movement extent of the trajectories 3 and 4. **We strongly recommend the readers to watch the provided videos in the supplementary file for a more direct understanding.**

Note that, in the I2V setting, both MotionCtrl and `CameraCtrl` are implemented on the save video diffusion model, SVD (Blattmann et al., 2023a), which excludes the influence stemming from the different base video generators. Thus, the better camera viewpoint controlling in the generated videos benefits from the better design choices of `CameraCtrl`.

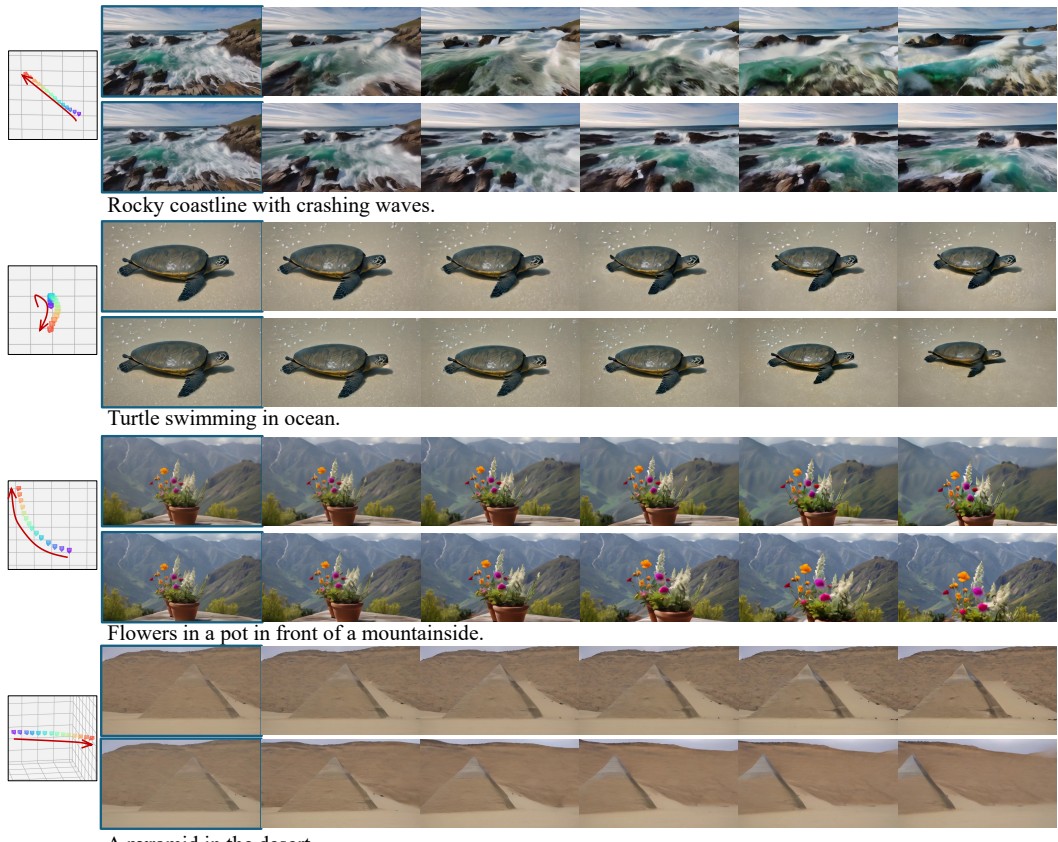

Rocky coastline with crashing waves.

Turtle swimming in ocean.

Flowers in a pot in front of a mountainside.

A pyramid in the desert.

Figure 10: **Qualitative comparison between MotionCtrl and `CameraCtrl` in I2V setting.** The condition images are shown in the first images of each row. These images are generated with the SDXL (Podell et al., 2023) taking the text prompts located below of every two rows as input. Note that, both MotionCtrl and `CameraCtrl` only condition on the conditioning images, not include the text prompts. The rows 1, 3, 5, and 7 are the results of MotionCtrl, while the results of `CameraCtrl` are in rows 2, 4, 6, and 8. Every two adjacent rows are generated with the same condition image and the same camera trajectory.

## H    MORE VISUALIZATION RESULTS

This section provides additional visualization results of `CameraCtrl`. Specifically, Appendix H.1 provides the various domain videos generated by integrating `CameraCtrl` with AnimateDiff (Guo et al., 2023b) in T2V setting. In Appendix H.2, we exhibit the generated videos of `CameraCtrl` in the I2V setting where the Stable Video Diffusion (SVD) (Blattmann et al., 2023a) is chosen as the base video generator. After that, video results of combining `CameraCtrl` with another video control method, SparseCtrl (Guo et al., 2023a) is shown in Appendix H.3. Finally, Appendix H.4 shows the flexibility of `CameraCtrl`.

### H.1    VISUALIZATION RESULTS OF VARIOUS DOMAIN T2V VIDEOS

**Visual results of RealEstate10K domain.** First, with the aforementioned image LoRA model trained on RealEstate10K dataset, and using captions and camera trajectories from RealEstate10K, `CameraCtrl` is capable of generating videos within the RealEstate10K domain. Results are shown in Fig. 11, the camera movement in generated videos closely follows the control camera poses, and the generated contents are also aligned with the text prompts.

**Visual results of original T2V model domain.** We choose the AnimateDiff V3 (Guo et al., 2023b) as our video generation base model, which is trained on the WebVid-10M dataset. Without the

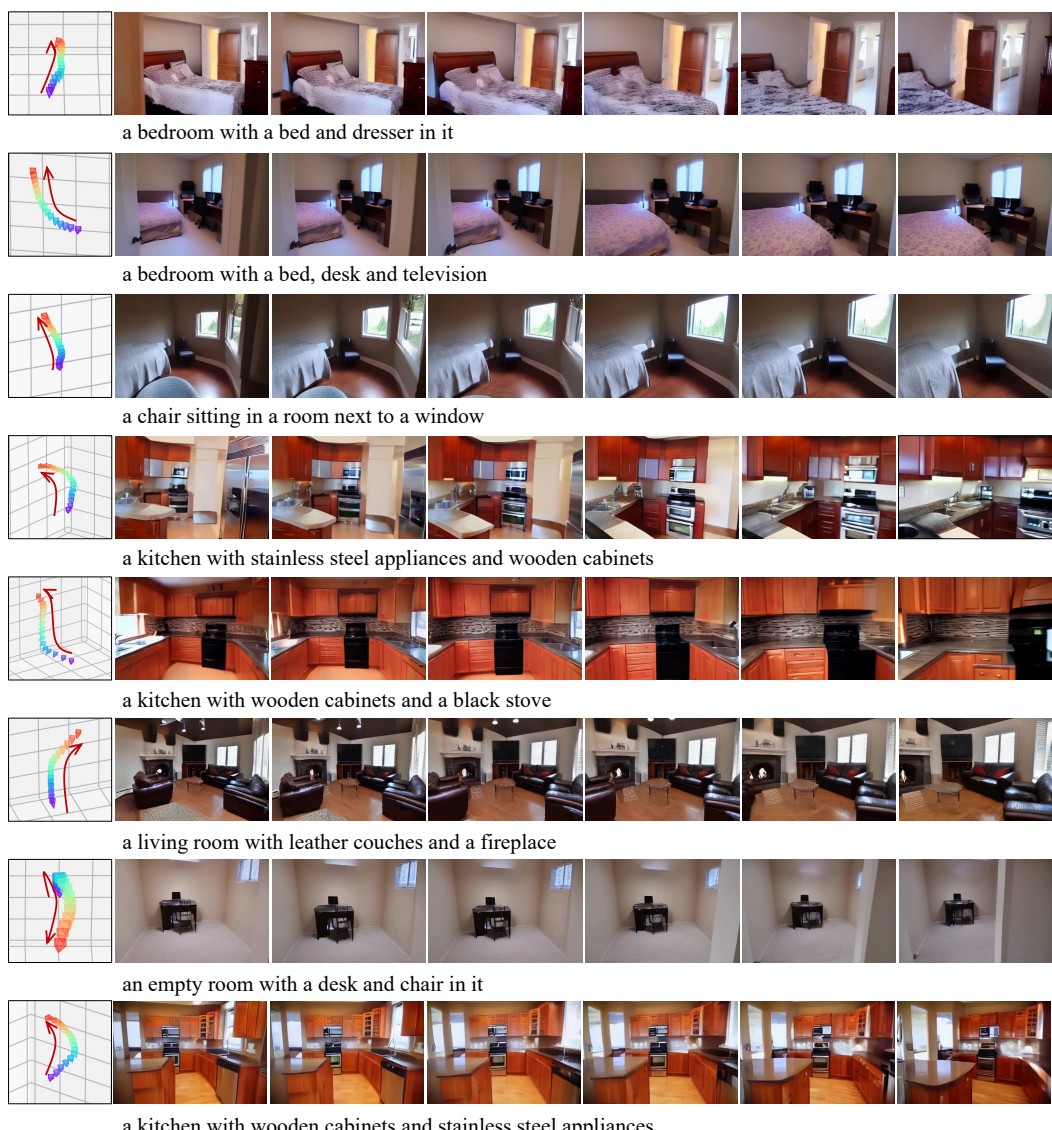

a bedroom with a bed and dresser in it

a bedroom with a bed, desk and television

a chair sitting in a room next to a window

a kitchen with stainless steel appliances and wooden cabinets

a kitchen with wooden cabinets and a black stove

a living room with leather couches and a fireplace

an empty room with a desk and chair in it

a kitchen with wooden cabinets and stainless steel appliances

Figure 11: **RealEstate10K visual results.** The video generation results of `CameraCtrl`. The control camera trajectories and captions are both from RealEstate10K test set.

RealEstate10K image LoRA, `CameraCtrl` can be used to control the camera poses during the video generation of natural objects and scenes. As shown in Fig. 12, with the same text prompts, taking different camera trajectories as input, `CameraCtrl` can generate almost the same scene, and closely follows the camera trajectories. Besides, Fig. 13 shows more visual results of natural objects and scenes.

**Visual results of some personalized video domain.** By replacing the image generator backbone of T2V model with some personalized generator, `CameraCtrl` can be used to control the camera poses in the personalized videos. With the personalized generator RealisticVision (civitai), Fig. 14 showcases the results of some stylized objects and scenes, like some uncommon color schemes in the landscape and coastline. Besides, with another personalized generator ToonYou (BradCatt), `CameraCtrl` can be used in the cartoon character video generation process. Some results are shown in Fig. 15. In both domains, the camera trajectories in the generated videos closely follow the control camera poses.

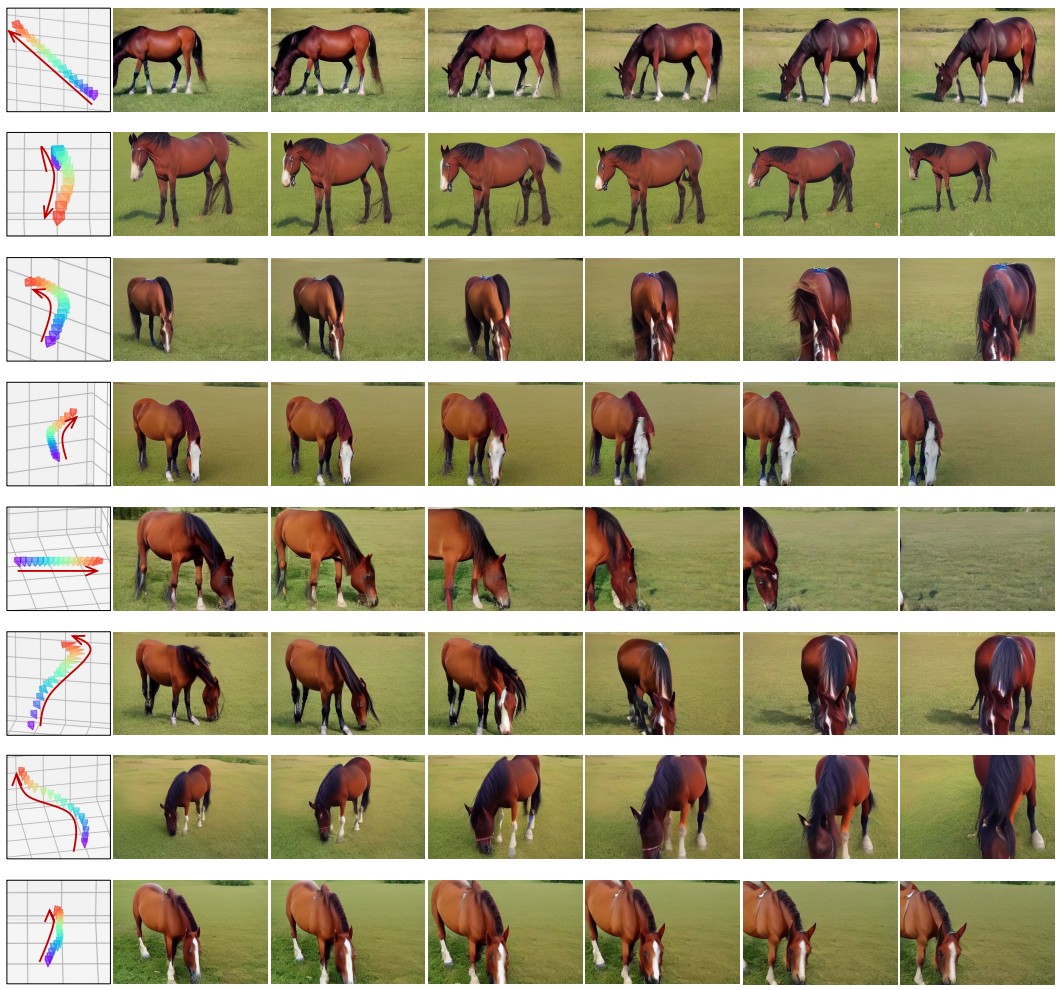

A horse is eating grass on the grassland.

Figure 12: **Using `CameraCtrl` on the same caption and different camera trajectories.** The camera control results of `CameraCtrl`. Camera trajectories are from RealEstate10K test set, all videos utilize the same text prompts.

## H.2 I2V VISUALIZATION RESULTS OF CAMERACTRL INTEGRATED WITH SVD

By taking the SVD as the base video generator to implement our `CameraCtrl`, we can sample videos with desired camera trajectories in the I2V setting. Fig. 16 shows some of them. The camera viewpoints of generated videos strictly follow the camera trajectory input, and video content also align with the condition images.

## H.3 INTEGRATING CAMERACTRL WITH OTHER VIDEO CONTROL METHOD

Fig. 17 gives some generated results by integrating the `CameraCtrl` with another video control method SparseCtrl (Guo et al., 2023a). The content of the generated videos follows the input RGB image or sketch map closely, while the camera trajectories of the videos also effectively align with the conditioned camera trajectories.

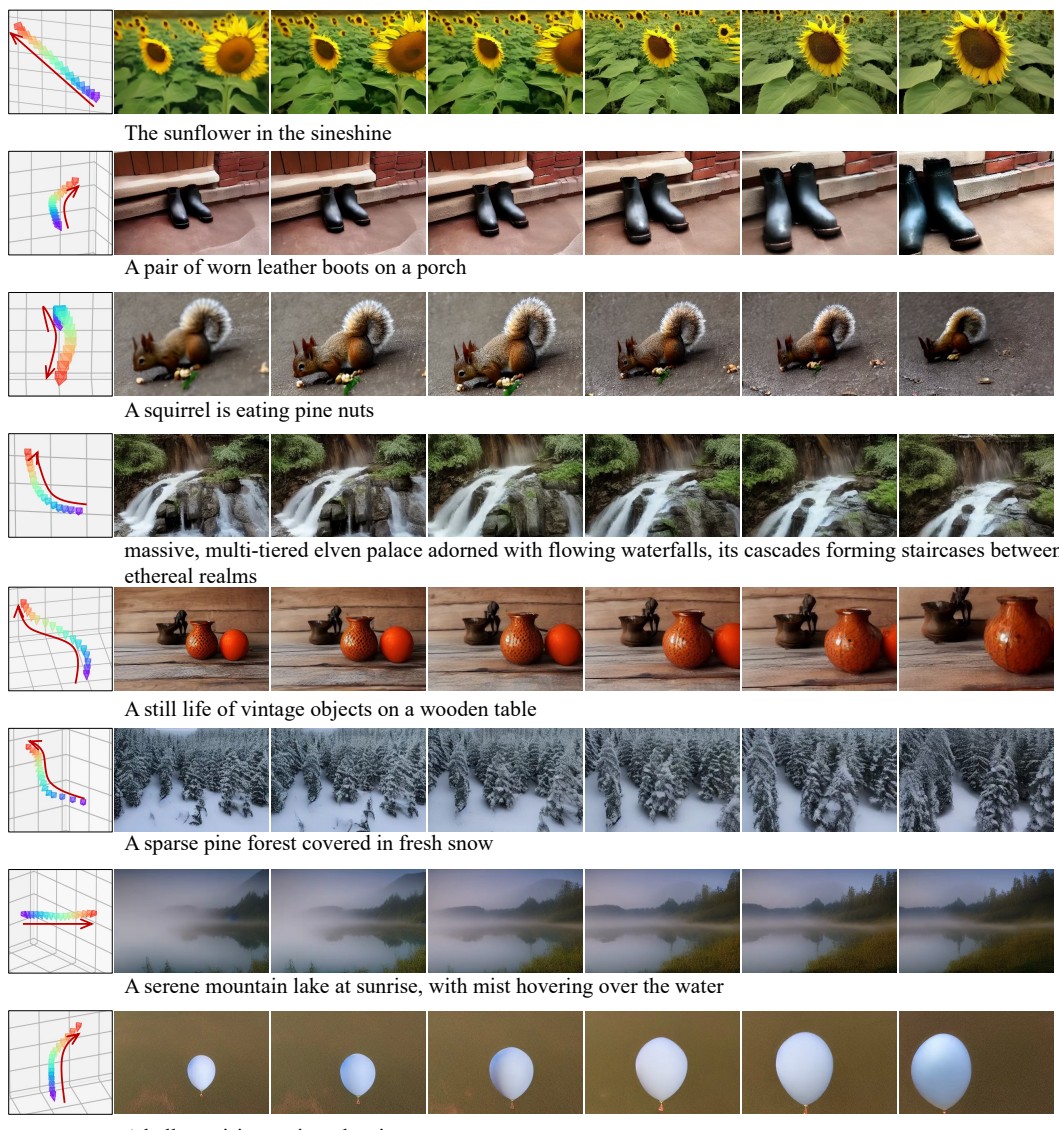

The sunflower in the sineshine

A pair of worn leather boots on a porch

A squirrel is eating pine nuts

massive, multi-tiered elven palace adorned with flowing waterfalls, its cascades forming staircases between ethereal realms

A still life of vintage objects on a wooden table

A sparse pine forest covered in fresh snow

A serene mountain lake at sunrise, with mist hovering over the water

A balloon rising up into the air

Figure 13: **Visual results of natural objects and scenes.** The natural video generation results of `CameraCtrl`. `CameraCtrl` can be used to control the camera poses during the video generation process of natural objects and scenes.

## H.4    FLEXIBILITY OF CAMERACTRL

**Different camera movement intensity.** By adjusting the interval between the translation vectors of two adjacent camera poses, we can control the overall intensity of the camera movement. As shown in the Fig. 18, we can make the camera movement more intense or more gradual.

**Controlling camera movement by adjusting intrinsic.** Since the Plücker embedding requires internal parameters during the computation, we can achieve camera movement by modifying the camera's intrinsic parameters. As shown in the Fig. 19, by changing the position of the camera's principal point (cx, cy), we can achieve camera translation (as shown in the first three rows). By adjusting the focal length (fx, fy), we can achieve a zoom-in and zoom-out effect, as shown in the last two rows.

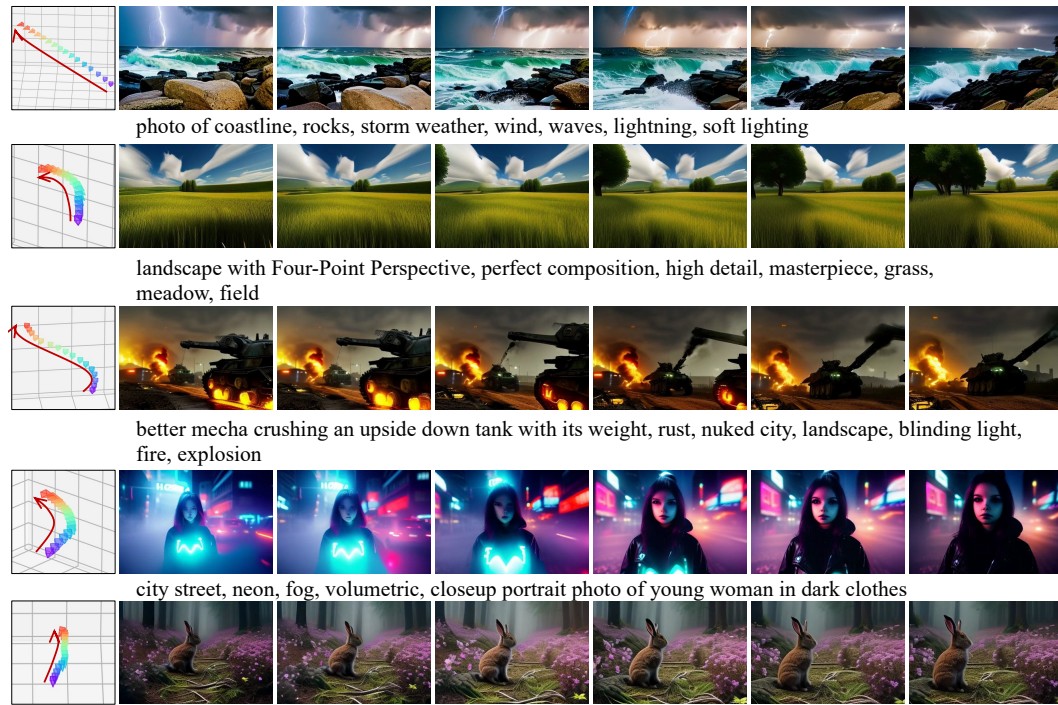

photo of coastline, rocks, storm weather, wind, waves, lightning, soft lighting

landscape with Four-Point Perspective, perfect composition, high detail, masterpiece, grass, meadow, field

better mecha crushing an upside down tank with its weight, rust, nuked city, landscape, blinding light, fire, explosion

city street, neon, fog, volumetric, closeup portrait photo of young woman in dark clothes

close up photo of a rabbit, forest, haze, halation, bloom, dramatic atmosphere, centred

Figure 14: **Visual results of stylized objects and scenes.** With the personalized generator RealisticVision (civitai), `CameraCtrl` can be used in the video generation process of stylized videos.

## I  FAILURE CASES

In Fig. 20, we provide some failure cases of `CameraCtrl`. The main problem for these cases is that when the rotation of the camera trajectory has a large extent, `CameraCtrl` cannot properly generate videos with enough rotation. The first and second rows take the vertical uniform rotation 100 degrees as input, but the generated videos cannot rotate 100 degrees. The same problem is kept for rows 3 and 4, where we desire a horizontal uniform of 150 degrees. However, there is only about a 90-degree rotation of the generated videos. The main reason for this failure situation may lie in that the training dataset (RealEstate10K) does not contain enough camera trajectories with a large degree of rotation. Thus, to improve the camera trajectory performance further, a dataset possessing a similar visual appearance to RealEstate10K and a larger camera pose distribution is needed.

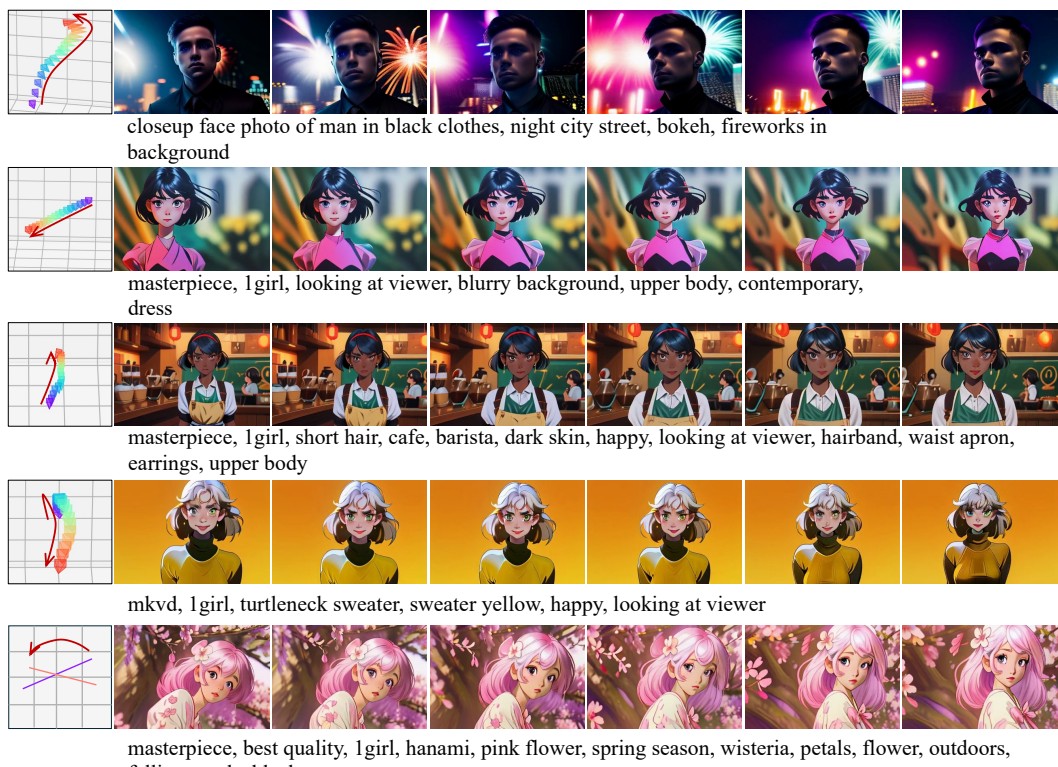

Figure 15: **Visual results of cartoon characters.** With the personalized generator ToonYou (Brad-Catt), `CameraCtrl` can be used in the video generation process of cartoon character videos.

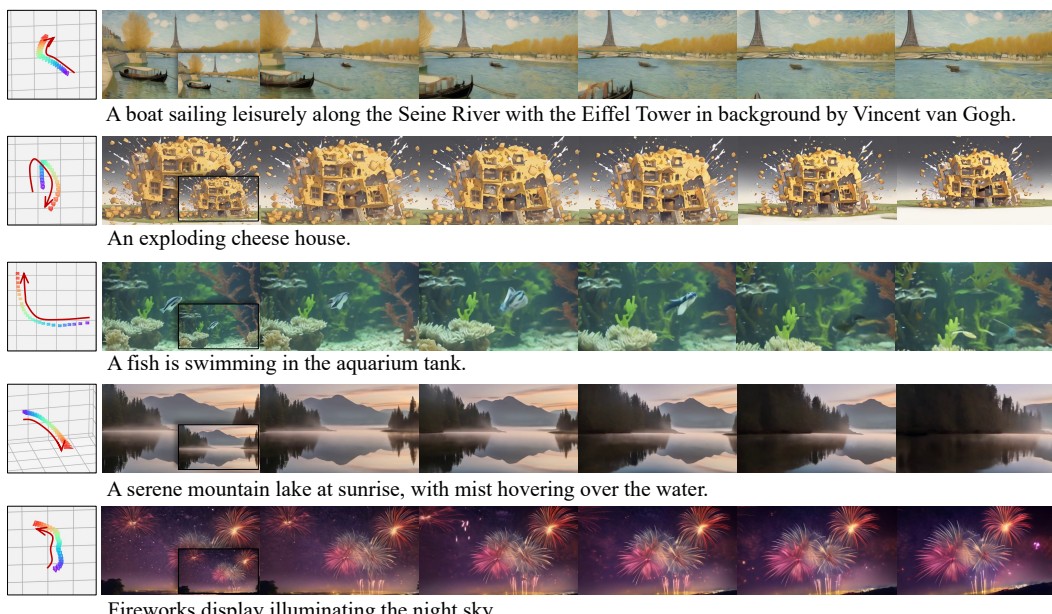

Figure 16: **Integrating `CameraCtrl` with SVD in the I2V setting.** The condition images are located in the right bottom corners of each rows first image. These images are generated by the text-to-image model SDXL with the text prompts down below each row as input. The condition signals of generated videos are only images, not include the text prompts.

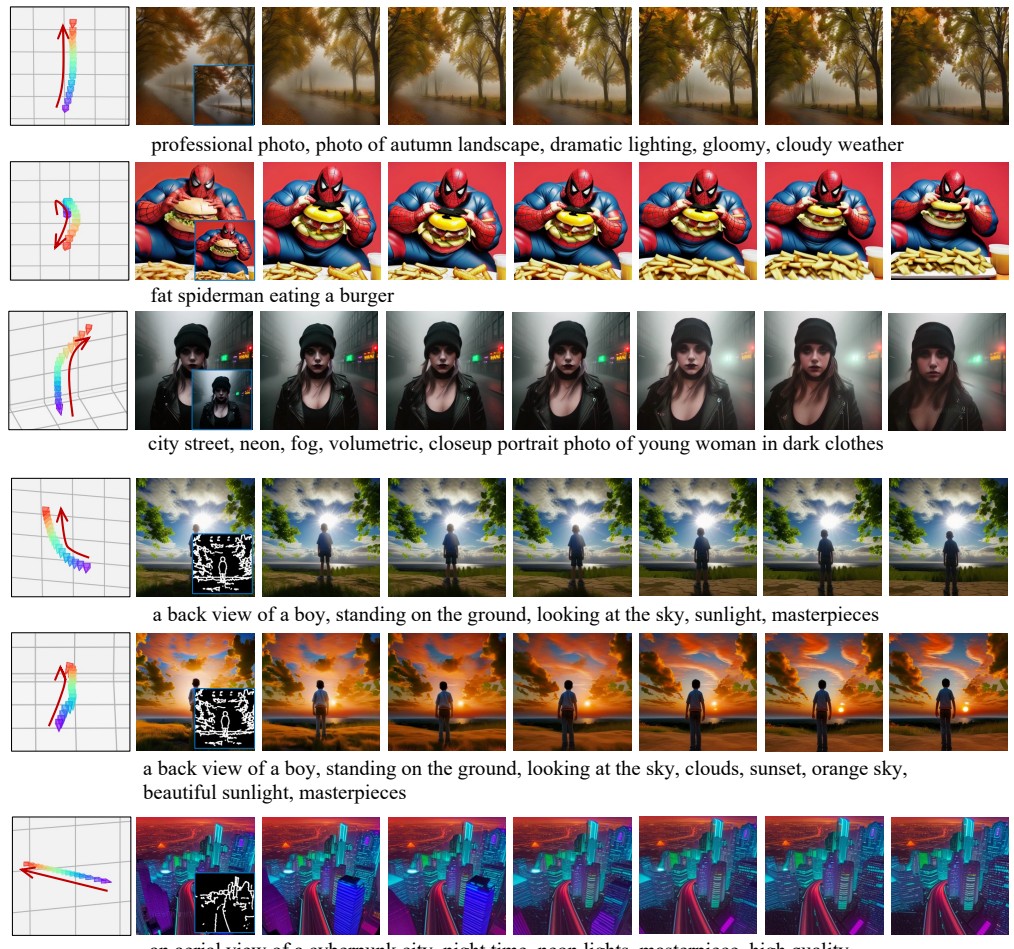

Figure 17: **Integrating `CameraCtrl` with other video generation control methods.** Row one to row three express the results by integrating the `CameraCtrl` with RGB encoder of SparseCtrl (Guo et al., 2023a), and row four to row six, shows videos produced with the sketch encoder of SparseCtrl. The condition RGB images and sketch maps are shown in the bottom right corners of the second images for each row. Note that, the camera trajectory of the last row is zoom-in.

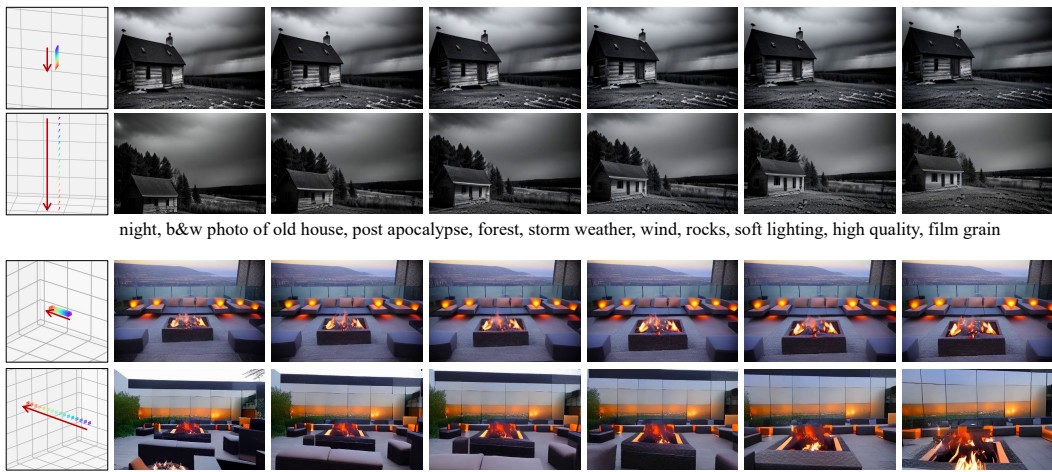

night, b&w photo of old house, post apocalypse, forest, storm weather, wind, rocks, soft lighting, high quality, film grain

an outdoor lounge area with a fire pit overlooking the city

Figure 18: **Camera movement intensity.** The first two rows taking the pan down camera trajectory as input, with the camera translation interval in the second row being four times that of the first row. The camera trajectory for the third and fourth rows are zoom in, with the camera translation interval in the fourth row being four times that of the third row.

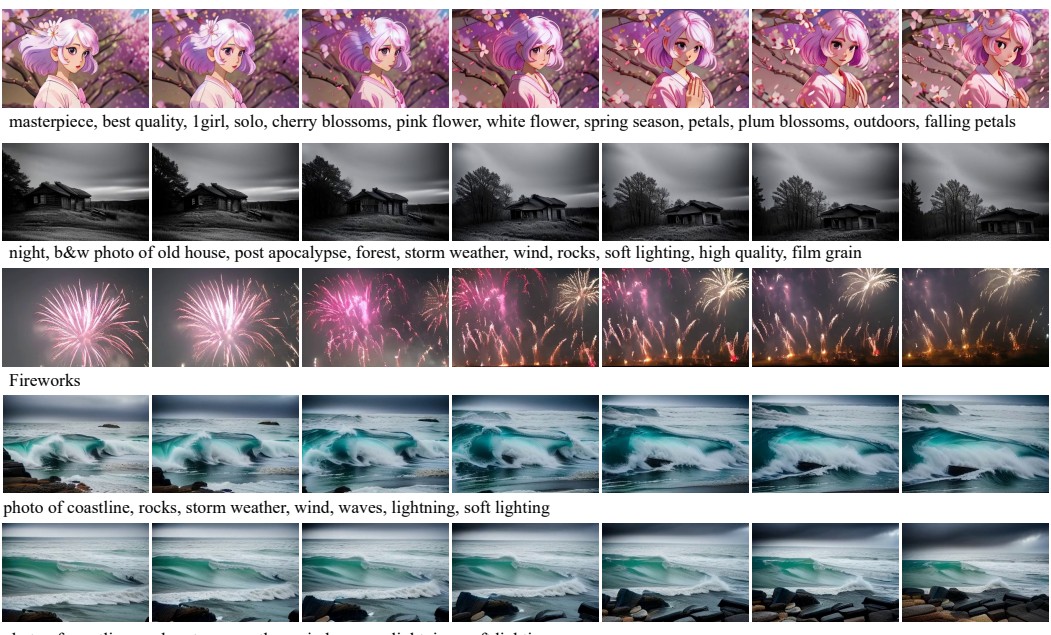

masterpiece, best quality, 1girl, solo, cherry blossoms, pink flower, white flower, spring season, petals, plum blossoms, outdoors, falling petals

night, b&w photo of old house, post apocalypse, forest, storm weather, wind, rocks, soft lighting, high quality, film grain

Fireworks

photo of coastline, rocks, storm weather, wind, waves, lightning, soft lighting

photo of coastline, rocks, storm weather, wind, waves, lightning, soft lighting

Figure 19: **Controlling camera movement by adjusting intrinsic.** The first three rows show the generated results using camera pan left, left up, right down, respectively. The last two rows take the zoom in, zoom out camera trajectories as the input. In each camera trajectory, all the camera poses have the same extrinsic matrix, the camera movement is implemented by adjusting the intrinsic parameters, cx and cy for the first three rows, fx and fy for the last two rows.

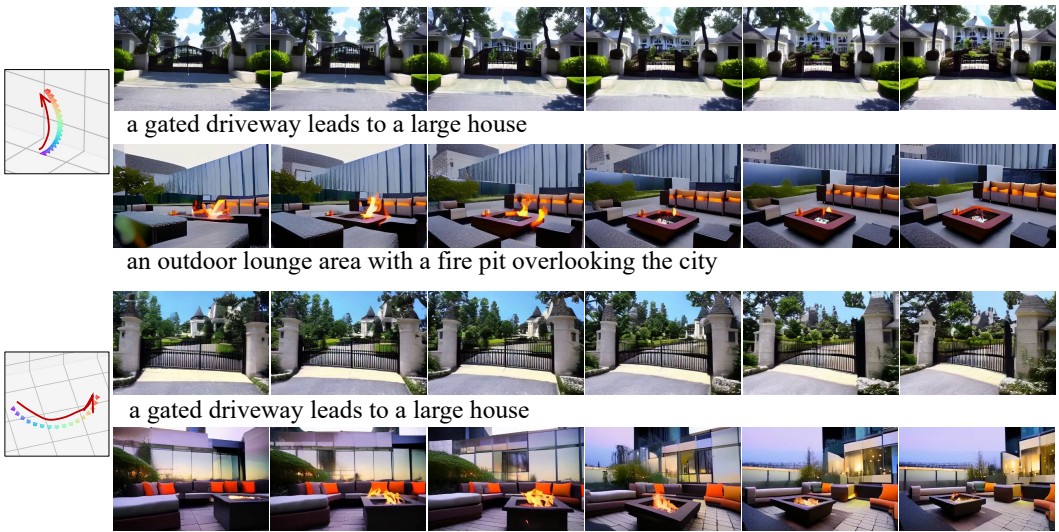

Figure 20: **Failure cases.** All results are generated with the `CameraCtrl` implemented on Animate-DiffV3 in the T2V setting. The camera trajectory for rows 1 and 2 is the vertical uniform rotation of 100 degrees. The trajectory horizontal uniform rotation 150 degrees is used during the generation of rows 3 and 4.

