# OpenReview forum: "CameraCtrl: Enabling Camera Control for Video Diffusion Models"
_ICLR.cc/2025/Conference — ICLR 2025 Poster_

### Official Review · Reviewer_UQ23 · 2024-10-21

**Soundness:** 2
**Presentation:** 3
**Contribution:** 3
**Rating:** 6
**Confidence:** 4

**Summary:**

The paper introduces a general-purpose encode for a moving camera that can be used to add camera conditioning to different video generator architectures. The paper considers three technical problems: representing the camera, injecting the representation in a video generator, and developing training data sufficient for learning the model. The camera encoding is based on Plucker rays, it is injected in a video generator via cross-attention layers added to the temporal self-attention of the video generator, and the training data combines synthetic and real-life datasets of static contest where the camera is known.

Empirically, the new module is integrated AnimateDiff and in and Stable Video Diffusion (SVD), two different video generators prompted using text. Compared to alternative like MotionCtrl, the approach has better metrics in terms of camera reconstruction and image quality.

**Strengths:**

* The problem considered by the paper is practically important: camera control is an important feature in many potential applications of video generation

* The approach is technically simple. The architecture design is reasonably well motivated. The performance is shown to be better than close competitors.

* Several applications beyond text-to-video generation with camera control are demonstrated, showing that the method successfully combines with other control types.

**Weaknesses:**

I remain unconvinced about the formulation of the problem (which is a limitation shared by prior works).

The paper aims at controlling camera motion accurately, but camera motion can only be interpreted in *relation* to the content generated in the video, and this relationship is not well defined in this work. This has several manifestations, discussed next.

First, there is the issue of scale: how a particular value for the camera translation should be interpreted depends on the scale of the generated scene, which is unknown and/or arbitrary. For example, a translation of one meter to our right is entirely negligible if we are looking at the sun, but if we are looking at the chair in front of us it is sufficient to take the object out of view entirely. There is no indication on whether the model is learning a "metric" (or canonical) interpretation of scale; this seems in fact unlikely given the nature of the training datasets which do not posses a well defined scale. The appendix briefly mentions scale as an issue of COLMAP, which requires scale normalisation for evaluation. But this is not a minor issue compared to the  much more serious concern that we cannot assume that the scale of the training data is know. Even if it has "some" value, the latter is often arbitrary and inconsistent (many real-life datasets like MVImageNet use COLMAP for camera annotation; synthetic ones like Objaverse have arbitrary scale).

Even if the training data was "metric", so that one could at least attempt to learn a meaningful canonical notion of scale, and thus interpret camera translation accordingly, this would still be difficult to do for a model. This is a reflection o the fact that very similar content can exist at different physical scales in the world (e.g., model of a city vs city), so learning  a canonical reference scale, insofar as one can be defined, is difficult.

Second, along with scale there is the issue of framing and aligning content and camera motion. Suppose you wanted to move the camera around an object. You could specify a camera trajectory that pivots around a given arbitrary physical point in front of you, in a turn-table-like fashion. Whether the network decides to place an object in correspondence of the pivot point, thus obtaining a turn-table-like effect, is not clear. Camera motion is only useful when its relation to the content is meaningful, so that the subject of the video is framed in the intended way.

Thirdly, there is the issue of the intrinsic parameters of the camera. It is not clear how these are handled in the paper. Specifically, the Plucker ray should be able to encode any camera model/focal length, but it is not obvious whether the authors train the model with different sets of cameras, and if the model is accurate enough to capture focal length correctly. Note that focal length would affect the apparent size of an object in an image, so this also has a strong effect in attempting to learn a canonical/metric interpretation of scale, which seems necessary in this formulation of the problem (see above).

Note that the issue of scale trickles down to evaluation. Even if the content of the video is not well matched to the camera motion (i.e., the framing and/or scale is not what one would expect), the model can still generate a video with the prescribed camera motion. The latter can then recovered using, e.g., COLMAP and compared to the input camera up to scale (for the translation component). The latter is described in appendix, but evaluation of camera reconstruction has been considered extensively int he literature; why not using one of the many standard scale-invariant metrics instead of proposing a new one (line 990)?

More worryingly, the authors state (line 983) that they manually discard badly generated videos where COLMAP fails in their evaluation (line 983). This can very well introduce a selection bias, where their method is favoured by simply discarding difficult videos more liberally than for others. While it is difficult to compare video generators directly due to the aleatoric nature of their task, nevertheless the authors should consider a metric that would at least remove this problem. For example, they could measure success rate, as in the percentage of generated videos where the camera is reconstructed at different levels of accuracy, generating corresponding curves. By prompting different models with the same textual prompts and the same camera motions, such curves would at least be more comparable.

The novelty of the paper is reasonable, but not outstanding. A very similar problem is considered in MotionCtrl and the paper is mostly an iteration that innovates on the three technical components discussed above (better camera encoding, better camera injection, and better data). There is nothing wrong with it, and the results are a bit better, but the conceptual innovation does not run very deep.

# Minor issues

* Line 195: the sentence structure is broken

* Line 460: Do you mean that COLMAP cannot reconstruct the generated videos? Is this an issue with COLAMP or with the quality of generation?

**Questions:**

* How is scale handled by your model? For example, what is the meaning/effect of translating the camera by, say, one unit to the right? What about two units?

* How do you make sense of the camera motion if you don't know what is going to be generated in the video?

* Does the choice of focal length (camera intrinsics) affect the generated video?

* Can you comment on the potential issues on the evaluation discussed above?

**Details Of Ethics Concerns:**

While, as noted by the authors, video generation is generally ethically sensitive, the delta in this paper is not likely to make video generation any worse or better than the prior works on which this paper is based on.

---

> ### Author Response · Authors · 2024-11-23
>
> **Weakness1 and Question1: How is scale handled by the CameraCtrl model? Can the model learn a “metric” interpretation of scale? what is meaning of translating the camera by one / two unit to the right?**
>
> **Response:** Our training data uses camera poses derived from COLMAP, which only captures relative scale that varies across different scenes. Thus, our model cannot learn metric-scale interpretation.
>
> **Regarding precise camera movements**: While we can't generate precise metric-based camera movements, we can control movement magnitude by adjusting the spacing between sequential camera poses. Figure 18 in our updated paper demonstrates this capability.
>
> **Weakness2 and question2: How you make sense of the camera motion without knowing what is to be generated?**
>
> **Response**: While exact content prediction of generated videos is not possible, users can guide the generation through text or image prompts to get reasonable expectations of the content. This helps users design suitable camera movements that align with anticipated videos. Besides, users can generate multiple videos with different camera paths and choose the one that best matches their creative vision.
>
> **Weakness3 and question3: How does the paper handle the camera intrinsic parameters? Does the choice of focal length affect the generated video?**
>
> **Response: First question:** Our approach incorporates camera intrinsic parameters in the Plücker embedding calculation (Equation 3 of paper).
>
> **Second question:** Focal length does influence the generated videos' camera movements. By adjusting the focal length, we can create zoom-in and zoom-out effects. These effects are demonstrated in Figure19 of updated paper.
>
> **Question4: Evaluation issues.**
>
> **Response: For question, why propose a new metric instead of using existing scale-invariant ones?:** CameraCtrl is one of the first works to introduce camera control in video generation, and we found no established benchmarks for evaluating camera alignment between generated videos and conditioning cameras. This led us to propose TransErr and RotErr metrics. Several concurrent works, like CamCo[1] and VD3D[2], adopted similar metrics. We welcome suggestions for scale-invariant metrics that could enhance our evaluation.
>
> **Regarding the removal of videos for RotErr and TransErr metrics**: COLMAP fails to estimate camera poses in two scenarios: when our base video generator produces outputs with low visual quality, and when the number of frames (16/25 in our setting) is too sparse for reliable camera pose reconstruction.
>
> In these cases, videos must be excluded from evaluation because COLMAP cannot reconstruct camera poses. Since low visual quality and sparse frames don't necessarily correlate with video complexity, this exclusion doesn't create selection bias.
>
> **A more appropriate metric:**
> The suggested success rate metric, while valuable, shares limitations with our current metrics. COLMAP only produces binary outcomes (success/failure), and once processed, accuracy measures (TransErr and RotErr) are fixed. We cannot adjust success rates without using entirely different video sets, making this metric face similar challenges as our current ones.
>
> Given these limitations, we avoid over-relying on TransErr and RotErr. Instead, we conducted a user study (Appendix E) comparing CameraCtrl with others, providing more meaningful validation of our method's effectiveness.
>
> **Question5: A very similar problem is considered in MotionCtrl.**
>
> **Response:** CameraCtrl, while addressing similar problem as MotionCtrl, offers key advantages:
> First, as a plug-and-play module, CameraCtrl seamlessly integrates with various image LoRAs for camera-controlled video generation, while MotionCtrl's fine-tuning training limits its generalization ability.
> Second, Our Plücker Embedding and ControlNet-like conditioning enable precise camera control, whereas MotionCtrl shows limitations in fine-grained camera movements, as proven in Figures 3, 9, and 10 of our paper.
>
> **Question6:  Line 195 has sentence structure broken.**
>
> **Response**: Thanks for pointing out, we have fixed it.
>
> **Question7: Line 460. Does the author means that COLMAP cannot reconstruct the generated videos? Is this an issue with COLMAP or the quality of generation?**
>
> **Response:** Yes, it is because the generated videos are of low quality. Our model builds upon base video generators, typically trained on complex datasets like WebVid. When training CameraCtrl on the Objaverse dataset (object-centric, simple background), the visual gap between datasets leads our model to learn some Objaverse-specific bias. When using RealEstate10K (scene-level dataset) prompts for evaluation, this results in poor visual quality. COLMAP fails to estimate camera poses due to this low quality.
>
> [1]Xu, Dejia, et al. CamCo: Camera-Controllable 3D-Consistent Image-to-Video Generation
>
> [2]Bahmani, Sherwin, et al. Vd3d: Taming large video diffusion transformers for 3d camera control

---

> ### Author Response · Authors · 2024-11-27
>
> Dear reviewer UQ23,
>
> Thank you for your time to provide the reviews for our paper. I want to remind you that I have submitted my rebuttal to your reviews,  specifically addressing the key concerns you raised regarding the scale handling, the camera trajectory and video content alignment, the usage and influence of camera intrinsics for CameraCtrl, the potential issues of evaluation, and the reason of COLMAP fail on videos generated by Objaverse-trained model.
>
> I have provided the detailed explanations and qualitative results to clarify these points and look forward to your feedback.
>
> Thank you for your time and consideration.

---

> > ### Comment · Reviewer_UQ23 · 2024-11-27
> >
> > > Response: Our training data uses camera poses derived from COLMAP [...] Figure 18 in our updated paper demonstrates this capability.
> >
> > Figure 18 does show some correlation between amount of translation and visual effect, meaning that the model learns implicitly a standardized notion of scale (assuming that this figure is representative). However, the figure also shows that the correlation is somewhat weak: the amount of motion in the top example is about five times larger in the second camera, but the visual effect does not seem to be quite as large, although this is difficult to say for certain.
> >
> > > Response: While exact content prediction of generated videos is not possible [...] best matches their creative vision.
> >
> > If the motion specification is understood in a qualitative way by the model in any case, why going through the effort to actually encoding the camera motion as a specific trajectory instead of a more qualitative description? (e.g., "pan to the left a bit" or "pan to the left a lot"). From an *application* perspective this is the control that a user can expect to get, given the weak correlation between specified trajectory and content.
> >
> > > Response: First question: Our approach incorporates camera intrinsic parameters in the Plücker embedding calculation (Equation 3 of paper).
> >
> > OK, thank you.
> >
> > > Focal length does influence the generated videos' camera movements. [...] Figure19 of updated paper.
> >
> > OK, interesting this seems to work well.
> >
> > > We welcome suggestions for scale-invariant metrics that could enhance our evaluation.
> >
> > Any paper that has looked at camera reconstruction up to scale would have such a metric. Just to cite one of many, SFMLearner uses ATE after estimating scale, or you can redefine ATE up to a similarity instead of a rigid transformation. You can start here:
> >
> > A benchmark for the evaluation of RGB-D SLAM systems. Sturm, Engelhard, Endres, Burgard, Cremers.
> > Proc.IROS, 2012.
> >
> > >  In these cases, videos must be excluded from evaluation because COLMAP cannot reconstruct camera poses. Since low visual quality and sparse frames don't necessarily correlate with video complexity, this exclusion doesn't create selection bias.
> >
> > The point is that COLMAP success can very well correlate with *video quality* (not so much video complexity), so you might be selecting for good quality generations, no?
> >
> > > The suggested success rate metric, while valuable, shares limitations with our current metrics. COLMAP only produces binary outcomes (success/failure), and once processed, accuracy measures (TransErr and RotErr) are fixed. We cannot adjust success rates without using entirely different video sets, making this metric face similar challenges as our current ones.
> >
> > I am not sure what the authors mean by "once processed, accuracy measures (TransErr and RotErr) are fixed". Here the point is to measure the rate at which generated videos are reconstructible by COMLAP, before also checking that the reconstructed trajectory matches the desired one. What is the problem with generating enough videos for this? What do the authors mean by a "entirely different video sets"? These are generated videos, they are always different no?
> >
> > > CameraCtrl [...] as proven in Figures 3, 9, and 10 of our paper.
> >
> > I accept that there is sufficient novelty compared to MotionCTRL

---

> > > ### Author Response · Authors · 2024-11-29
> > > **(1/2) Response to reviewer UQ23's comments**
> > >
> > > Thank you for your constructive feedback and the increased score. We appreciate your thoughtful suggestions and would like to response to some of your comments as follows.
> > >
> > > > However, the figure also shows that the correlation is somewhat weak: [...] but the visual effect does not seem to be quite as large, although this is difficult to say for certain.
> > >
> > > **Response**: For the top case in Figure18, the translation intervals in the first row between every two consecutive frames is four times of those in the second row. This can be clearly observed in our supplementary video materials, which provide reviewers with an intuitive visualization of the motion differences between sequences. While keeping high-quality results, the camera motion in the second row already reaches a considerable magnitude.
> > >
> > > > Why going through the effort to actually encoding the camera motion as a specific trajectory instead of a more qualitative description? (e.g., "pan to the left a bit" or "pan to the left a lot").
> > >
> > > **Response**: CameraCtrl aims to achieve precise and diverse camera viewpoint control in video generation. To realize this goal, we integrate camera pose-related information into the video generation models. While qualitative text descriptions (e.g., "pan left slightly") offer an alternative control method, they are insufficient for specifying complex camera motions with high precision.
> > >
> > > > Any paper that has looked at camera reconstruction up to scale would have such a metric. [...] ATE up to a similarity instead of a rigid transformation.
> > >
> > > **Response**: Thank you for recommending ATE. The absolute trajectory error (ATE) methodology provides a more established and widely accepted approach for trajectory alignment than our method, particularly in handling scale issues and coordinate system alignment.
> > >
> > > Both ATE and our method need to address two inherent issues in COLMAP camera pose estimation: scale ambiguity (since COLMAP reconstructions are only accurate up to scale) and arbitrary coordinate system alignment (since COLMAP reconstructs camera poses in its own coordinate system, which may differ from the ground truth coordinate system through arbitrary rotation and translation). The main difference between ours and ATE’s lies in the alignment approach. While our method uses the first frame as reference point for coordinate alignment and the first two frames for scale adjustment, ATE employs global optimization across all frames for both scale and coordinate system alignment, potentially achieving more robust results by considering the complete trajectory. After the alignment, the calculation of TransErr and RotErr shares a similar philosophy between our method and ATE's.
> > >
> > > Following your suggestion, we have modified our evaluation pipeline. We adopt the ATE’s method to deal with the scale ambiguity and arbitrary coordinate system issues for the alignment of ground truth poses and estimated poses. Then, we calculate the RotErr and TransErr between the aligned estimated poses and ground truth poses using the equations 4 and 5 in paper. The results of CameraCtrl and MotionCtrl are in the following table. Our CameraCtrl still better than MotionCtrl.
> > >
> > > Table 1. Quantitative comparison between CameraCtrl and MotionCtrl using the modified TransErr and RotErr. Both methods use the Stable Video Diffusion as the base video generator.
> > >
> > > | Method | TransErr ↓ | RotErr ↓ |
> > > |---------|------------|-----------|
> > > | MotionCtrlSVD | 9.21 | 1.25 |
> > > | CamerCtrlSVD | 8.42 | 1.11 |
> > >
> > > > so you might be selecting for good quality generations, no?
> > >
> > > **Response**: We selected high-quality videos because COLMAP is unable to estimate camera poses from low-quality ones. The visual quality is determined by the underlying video generators, and importantly, remains independent of camera trajectory complexity. This enables evaluation across both simple and complex camera motions. Since both CamerCtrl and MotionCtrl share the same base video generator, our video selection process ensures a fair comparison of TransErr and RotErr metrics between methods.
> > >
> > > > I am not sure what the authors mean by "once processed, accuracy measures (TransErr and RotErr) are fixed". What do the authors mean by a "entirely different video sets"?
> > >
> > > **Response**: A more detailed description of "once processed, accuracy measures (TransErr and RotErr) are fixed" is as follows:  When processing a set of videos through COLMAP, not all yield successful camera pose estimations. Our metrics (TransErr and RotErr) are calculated only using the successful estimations compared against their ground truth poses. Due to COLMAP's deterministic nature, running the same video set multiple times will consistently produce identical results - same TransErr, RotErr, and success rates.
> > >
> > > Therefore, to evaluate different success rates, different video sets must be used, as each video's success/failure outcome is fixed.

---

> > > ### Author Response · Authors · 2024-11-29
> > > **(2/2) Response to reviewer UQ23's comments**
> > >
> > > >Here the point is to measure the rate at which generated videos are reconstructible by COMLAP, before also checking that the reconstructed trajectory matches the desired one. What is the problem with generating enough videos for this?
> > >
> > > **Response**: There is no problem with generating enough videos. However, let us address the context of our discussion regarding success rates. The reviewer suggests that success rate (*as in the percentage of generated videos where the camera is reconstructed at different levels of accuracy*) would be a better metric. While we believe it shares limitations with our current metrics for the following reasons:
> > >
> > > COLMAP failures primarily stem from video quality, not trajectory complexity. Success rates thus mainly indicate the proportion of high-quality videos rather than camera control performance.
> > >
> > > Our metrics are calculated by:First computing errors for each camera pose pair (Equations 4 and 5). Then averaging across successful videos. These metrics are independent of success rates. For example, with 1000 videos and 800 successful case, we can:(1)Reporting 100% success rate (800 successful videos only); (2)Reporting 80% success rate (all 1000 videos). Both scenarios yield identical TransErr and RotErr values.
> > >
> > >  Therefore, the reviewer's suggested success rate-TransErr (RotErr) curve would not provide additional insights beyond what TransErr and RotErr alone already reveal.

---

> > > ### Author Response · Authors · 2024-12-02
> > >
> > > Dear reviewer UQ23,
> > >
> > > As the deadline of the author-reviewer discussion period is approaching, could you please take a look at our newly added comments to your comments (if not yet) and see if it addressed you concerns or you have any other questions?
> > >
> > > Best,
> > >
> > > Authors

---

> ### Comment · Reviewer_UQ23 · 2024-12-02
> **Further comments on the response**
>
> As you have noticed, I did  upgraded my score due to the other answers, but I would still argue this point:
>
> > "COLMAP failures primarily stem from video quality, not trajectory complexity. Success rates thus mainly indicate the proportion of high-quality videos rather than camera control performance."
>
> A good video generator will have to balance capabilities such as control, quality, fidelity, and yield. For example:
>
> - A generator may afford perfect camera control, but generate a single default scene ignoring other conditionings. This would be useless (poor fidelity).
>
> - A generator may afford perfect camera control, but only work for 10% of the input prompts (90% of the videos would be discarded as "failed" or "poor quality). This generator would be poor as it would fail for most prompts (poor yield).
>
> - A generator may outperform another on camera control, but only on 10% of "high quality" videos, and lose to the other on 90% of, say, "medium quality" videos. Hence, one model would be better in the "high quality" regime, but the other would be better in most cases.
>
> In short, "acing" only one aspect/metric does not imply that the overall result is satisfactory or even useful. It is entirely possible that by  optimising for one metric (e.g., camera control) one would hurt others (e.g., yield, fidelity, quality).

---

> > ### Author Response · Authors · 2024-12-02
> >
> > Thanks for your reply, we have noticed you upgraded score, and greatly appreciate of that.
> >
> > >A good video generator will have to balance capabilities such as control, quality, fidelity, and yield.
> >
> > We think the comprehensive metrics (FVD for evaluating fidelity, CLIPSIM for prompt alignment, FC for frame consistency, and ODD for object dynamic degree) used in our paper can demonstrated compared with other method, CameraCtrl can achieve better or comparable results and has better camera viewpoint control ability.
> >
> > We thank again for the discussions with you during this period, your insightful feedback has been invaluable, inspiring improvements in both CameraCtrl and our potential researches.

---

### Official Review · Reviewer_BKUF · 2024-10-26

**Soundness:** 3
**Presentation:** 4
**Contribution:** 3
**Rating:** 8
**Confidence:** 4

**Summary:**

The authors explore the problem of conditioning a pretrained video generation model on a desired camera motion. The problem is of high relevance not only to the video generation community, but also in 3D and 4D generation. The authors propose the use of a frozen video generation backbone, coupled with a camera encoder block, injecting camera information into the backbone. The method is simple, sound, clearly written and accompanied by a comprehensive evaluation validating the importance of each of the main components: camera embeddings, encoder architecture, feature injection, and training datasets.

**Strengths:**

- The problem treated in the paper is of high significance to video generation, and 3D/4D generation.
- The paper is very clearly written. Every section of the paper was easy to follow and it appears the information in the paper would be sufficient to replicate the presented method and results. In addition, code is available.
- I appreciate the simplicity of the method. The authors identified the most crucial components to enable camera control: use of Plucker embeddings, camera encoder architecture, how to couple the encoder to the retrained backbone, and training data recipe. Without adding bells and whistles, the authors show good camera control performance.
- Evaluation is solid and informative. Every of the most critical design dimensions (use of Plucker embeddings, camera encoder architecture, how to couple the encoder to the retrained backbone, and training data recipe) is validated in a respective ablation, providing useful insights. The model is demonstrated on diverse video generation backbones
- Qualitative evaluation is rich. Limitations are shown and highlighted

**Weaknesses:**

- Discussion of the camera representation could be strengthened by an accompanying figure.
- While Plucker embeddings were found more effective than a set of straightforward alternative representations, their properties do not appear to be completely aligned to the nature of the proposed approach. See questions.

**Questions:**

- The choice of the Plucker representation appears effective when compared to Euler angles, quaternions and raw values, though some properties of the Plucker representation appear suboptimal for the purpose of achieving camera control. In particular, the set of points along a certain camera ray will have a constant embedding. Sitzmann (2021) introduces the use of Plucker coordinates in the context of LFNs, for which this property is well justified and beneficial. In the context of the current paper, though, Plucker embeddings appear to obscure the camera origin, a fundamental information for the model. With each Plucker embedding defining a set of points along a ray, the only way for the model to recover the camera origin is to solve a system of equations involving multiple rays. This is a necessary, but likely difficult task for the model to perform. Why is this property desirable for this work? Did the authors experiment with alternative representations not suffering from this drawback such as (o, d) or (normalized_o, d) (for some choice of normalization for o) which would make the camera origin readily available to the model?

---

> ### Author Response · Authors · 2024-11-23
>
> **Weakness1: The discussion of camera representation could be strengthened by a figure.**
>
> **Response**: Thanks for your suggestion. We have added Figure 6 in the Appendix of the updated paper to illustrate different camera representations.
>
> Additionally, we provide Figure 7 in the Appendix to compare video results generated using three camera representations: numerical values of camera matrices, ray directions with camera origin (see next response), and Plücker embedding.
>
> Using a forward-moving camera trajectory with a final rightward shift, we demonstrate that: raw matrix values fail to capture the rightward movement, ray directions with camera origin create abrupt shifts in final frames, while Plücker embedding achieves smooth transitions throughout, including the final rightward movement. These results further verify the effectiveness of using Plücker embedding as the camera representation.
>
> **Question1: The set of points along a certain camera ray will have a constant Plücker embedding and Plücker embedding cannot explicitly provide the camera origin information, need solving equations, why is this property desirable for CameraCtrl? Did the author using another representation, that could make the camera origin readily available to the model?**
>
> **Response: For the first question:**
>
> The Plücker embedding parameterizes camera poses as a spatial feature where points along each ray share the same embedding. This representation provides the model with a richer space along camera rays, enabling better implicit depth adjustment for complex scene modeling in video generation.
>
> Using (o, d) or (o, normalized_d) for camera representation limits the model's capacity to represent complex scenes, as this representation only encodes points with unit length for each ray.
>
> While camera origin is crucial for describing camera location, neural network-based implicit scene representations like NeRF demonstrate that 3D scenes can be represented using only sampled points from camera rays without explicit camera origin information. This insight inspired our implicit conditioning mechanism for video diffusion models. To verify this approach, we compared it against a camera representation using (o, d) with explicit camera origin conditioning.
>
> As shown in the following table (also in the updated paper Table 2(a)), the explicit camera origin approach underperformed compared to Plücker embedding across all metrics (FVD, RotErr, and TransErr). This demonstrates that even without explicit camera origin, the model can effectively leverage Plücker embedding to infer and condition camera poses for video generation.
>
> Table 1: **Ablation study on camera representations**
>
> | Representation type | FVD↓ | TransErr↓ | RotErr↓ |
> |-------------------|------|-----------|----------|
> | Ray direction + camera origin | 232.3 | 13.21 | 1.57 |
> | Plücker embedding | **222.1** | **12.98** | **1.29** |

---

> ### Comment · Reviewer_BKUF · 2024-11-23
>
> I thank the authors for taking the time to answer to my concerns with additional quantitative evaluation. This fully addresses my concerns.
> After carefully reading other reviewer's comments, I continue to think that the paper is significant to the video generation and 3D-4D generation domains, is well written, and contains solid and informative qualitative and quantitative evaluation.
> I continue to argue for its acceptance.

---

> > ### Author Response · Authors · 2024-11-25
> >
> > We thank you for your time to read our rebuttal, recognize our additional quantitative results, and give us the prompt feedback. We appreciate your careful consideration of both our work and the other reviewers' comments, and your continued support for the paper's acceptance. Your feedback throughout this discussion process has been valuable in helping us strengthen our presentation and evaluation.
> >
> > Thank you again for your time and constructive engagement with our work.

---

### Official Review · Reviewer_7FvC · 2024-11-02

**Soundness:** 3
**Presentation:** 4
**Contribution:** 3
**Rating:** 6
**Confidence:** 4

**Summary:**

This paper proposes a plug-and-play module for control of camera pose. This module can be integrated to various video diffusion models. This module leverages Plücker embeddings to encode camera movements, offering detailed spatial representation compared to traditional numerical values (e.g., extrinsic matrix).

**Strengths:**

* This is one of the early works that enables adding camera control to existing video models. I believe it will have a positive impact on various downstream tasks.
* While the way of achieving camera control itself isn't technically very novel, the paper presents an intuitive design and provides extensive experimental comparisons on various conceivable design choices (e.g., where to inject the control and different camera representations).
* It is interesting that camera control can be achieved with a lightweight additional adapter without further tuning the video model.

**Weaknesses:**

* Since it is fine-tuned on static scene data, it tends to show somewhat reduced ability in generating dynamic motions compared to the original video diffusion baseline (as shown in ODD). But, it shows better performance compared to other methods.
* In object-centric video generation, it appears that the movement of the object is minimal compared to the background. Ideally, models fine-tuned on Objaverse or MVImgNet should be able to generate camera trajectories covering 360 degrees of the object. I'm curious if this model can effectively generate videos when provided with such camera trajectories as input (e.g., generating a video from a frame showing the front of a person to one showing the back).

**Questions:**

* Why are static scene video datasets mainly used without general video datasets? (For example, one could use general video datasets for training along with estimated camera trajectories from various models like particleSFM.)
* This didn't affect my rating, but I'm curious about the authors' opinions on whether this method can be generally applied to DiT-based methods that do not separate temporal and spatial aggregation.

---

> ### Author Response · Authors · 2024-11-23
>
> **Weakness1: Since CameraCtrl finetuned on static dataset, the base video generators’ ability of generating dynamic videos may be reduced**
>
> **Response:** During training, we strategically freeze the temporal layers of the base video generator while only fine-tuning the camera module to preserve the learned motion prior in video generative models as much as possible. Thanks to this design choice, our model doesn't harm the dynamics of the base model. As shown in following table, CameraCtrl achieve the comparable ODD score with the origianl video diffusion model.
>
> Table.1 **Quantitative comparisons on ODD between base video generation models and CameraCtrl.** CameraCtrlSVD and MotionCtrlSVD represent CameraCtrl and MotionCtrl implemented with the Stable Video Diffusion as base video generation model.
>
> | Method | Object Dynamic Degree |
> |---------|---------------------|
> | Stable Video Diffusion | 47.5 |
> | MotionCtrlsvd | 41.8 |
> | CameraCtrlsvd | 46.5 |
>
> **Weakness2: In object-centric video, the movements of foreground objects is less than the background. If train CamerCtrl on Objaverse or MVImageNet, can we generate an object-centric video with larger camera movement around the foreground?**
>
> **Response:** Yes! For our experiments training the camera control model on Objaverse, we successfully generated object-centric videos featuring 360-degree horizontal rotations around objects, achieving results comparable to those in the Objaverse dataset.
>
> **Question1: Why are the static scene video datasets instead of dynamic video datasets used for training CameraCtrl?**
>
> **Response:**
> Our goal is to develop precise camera control for video generative models, which requires datasets with diverse, extensive camera motions and accurate camera annotations.
>
> Dynamic scene datasets like WebVid typically feature minimal camera movement. Additionally, current structure-from-motion techniques struggle with dynamic objects, making it challenging to obtain accurate camera annotations for dynamic videos.
>
> Therefore, we chose static datasets such as RealEstate10K, Objaverse, and MVImageNet, which provide calibrated camera poses, to study camera conditioning in video generation. In future work, we plan to leverage both static and dynamic videos for model training: static videos for camera module learning and dynamic videos for temporal dynamics.
>
> **Question2: Whether the proposed method can be generally applied to DiT-based methods that do not separate temporal and spatial aggregation?**
>
> **Response**: Yes! Our method could be adapted for use with DiT-based models that do not explicitly separate temporal and spatial aggregation. The core concepts of our method are as two folds: (1) using the Plücker embedding to provide a pixel-wise spatial embedding, that can provide accurate geometric interpretation for each pixel in a video; (2) freezing the base video generation model to utilize its dynamics and versatile content generation ability.  Both concepts can be seamlessly integrated into the DiT-based video generation models. For the potential issue of “do not separate temporal spatial aggregation”, we can also do the spatial temporal modeling in the camera encoder. For example, we can tokenize the Plücker embedding spatial-temporally, process these tokens through a lightweight encoder, and combine them with the spatiotemporal features of diffusion latents.

---

> ### Author Response · Authors · 2024-11-27
>
> Dear reviewer 7FvC,
>
> Thank you for your time to provide the reviews for our paper. I want to remind you that I have submitted my rebuttal to your reviews,  specifically addressing the key concerns you raised regarding the potential degeneration of object motion, the object centric videos with 360 degree camera motion, the reason of using static video dataset, and the compatibility of CameraCtrl on DiT-based video generation models.
>
> I have provided the detailed explanations and quantitative results to clarify these points and look forward to your feedback.
>
> Thank you for your time and consideration.

---

> ### Author Response · Authors · 2024-12-02
>
> Dear reviewer 7FvC,
>
> Thank you for your time to provide the reviews and support regarding our paper's acceptance. As the deadline of the author-reviewer discussion period is approaching, could you please take a look at our rebuttal (if not yet) and see if it addressed you concerns or you have any other questions?
>
> Best,
>
> Authors

---

### Official Review · Reviewer_FegY · 2024-11-05

**Soundness:** 2
**Presentation:** 2
**Contribution:** 3
**Rating:** 6
**Confidence:** 4

**Summary:**

CameraCtrl enhances video generation by enabling precise camera pose control in video diffusion models, preserving narrative nuances through cinematic language. This approach integrates a plug-and-play camera pose control module and demonstrates improved controllability and generalization, with experimental results showcasing its effectiveness across different models.

**Strengths:**

The motivation is clear.

The results appear promising and solid.

The experiments are thorough.

The writing is easy to follow.

**Weaknesses:**

Applying a controlnet-like network to incorporate control signals into video diffusion isn’t particularly novel, though the use of Plücker embeddings adds a degree of originality.

Since the RealEstate10K dataset mainly features static videos with few moving objects, could this lack of motion introduce a bias that undermines the base model’s natural grasp of object motion? The foreground objects in the examples provided show limited movement.

There is a need for comparisons and discussion with recent camera control methods, such as VD3D.

**Questions:**

Please see the weakness part.

---

> ### Author Response · Authors · 2024-11-23
>
> **Question1: ControlNet-like network is less novel, though using Plücker embedding adds originality.**
>
> **Response**:  Unlike previous methods that directly use numerical camera parameters, our choice of adopting the Plücker embedding (a pixel-wise spatial embedding) as camera representation enables geometric interpretation for every pixel for detailed spatial modeling of generated videos. For this pixel-wise spatial camera representation, using a ControlNet-like encoder to inject the camera control signal is a natural choice, facilitating the precise modeling of spatial structure regarding the conditioned cameras. This combination enables CameraCtrl to achieve fine-grained control over camera movement in video generation, distinguishing our approach from prior work.
>
> **Question2: Could the static RealEstate10K dataset introduce a bias that undermines the base model’s natural grasp of object motion?**
>
> **Response**: The temporal dynamics are modeled by the temporal attention layer of the base video generation models. During training, we freeze all parameters of the base video generation model and only train the camera module. This approach helps maintain motion diversity in generated videos by preserving the motion prior encoded in the temporal layers.
>
> As shown in the following table, our model achieves comparable ODD scores with the base video generation model and outperforms other methods that do not freeze the temporal attention layer. These results demonstrate our model's strong capability in generating dynamic video content.
>
> In the future, we plan to improve our method by incorporating more dynamic videos with camera annotations. We will use dynamic structure-from-motion to obtain camera poses, then jointly train CameraCtrl on both static and dynamic videos like CVD[3] and Cavia[5].
>
> Table.1 **Quantitative comparisons on ODD between base video generation models and CameraCtrl.** CameraCtrlSVD and MotionCtrlSVD represent CameraCtrl and MotionCtrl implemented with the Stable Video Diffusion as base video generation model.
>
> | Method | Object Dynamic Degree |
> |---------|---------------------|
> | Stable Video Diffusion | 47.5 |
> | MotionCtrlsvd | 41.8 |
> | CameraCtrlsvd | 46.5 |
>
> **Question3: Need the comparisons and discussion with the recent camera control methods, like VD3D.**
>
> **Response**: We have added the discussions between CameraCtrl and recent camera control methods in the Appendix.B of the updated paper. It is also put as follows:
>
> Recent works have explored camera control in video generation, addressing different aspects of the challenge. VD3D[1] integrates camera control into a DiT-based model with a novel camera representation module in spatiotemporal transformers. CamCo[2] leverages epipolar constraints for 3D consistency in image-to-video generation. CVD[3] uses the camera control method and extends it to enable multi-view video generation with cross-view consistency. Recaprture [4] enables video-to-video camera control, effectively modifying viewpoints in existing content. However, it's limited to simpler scenes and struggles with complex or dynamic environments. Cavia[5] enhances multi-view generation through training on diverse datasets, improving cross-view consistency. The paper CMG[6] improves camera control accuracy using a classifier-free guidance-like mechanism in a DiT-based model. Despite the numerous works addressing camera control in the video generation process, to our best knowledge, CameraCtrl is a very early method to achieve accurate camera control in video generation models. It provides valuable insights and a solid foundation for future advancements in related fields, such as video generation, as well as 3D and 4D content generation.
>
> **Regarding quantitative comparison**: Since these papers have not released their source code and use different evaluation videos for TransErr and RotErr metrics, we cannot refer to their reported numbers in papers for a fair comparison across all methods. We will update our comparison results using the same set of evaluation videos once their implementations are made public.
>
> [1] Bahmani, Sherwin, et al. "Vd3d: Taming large video diffusion transformers for 3d camera control." arXiv:2407.12781 (2024).
>
> [2] Xu, Dejia, et al. "CamCo: Camera-Controllable 3D-Consistent Image-to-Video Generation." arXiv:2406.02509(2024).
>
> [3] Kuang, Zhengfei, et al. "Collaborative Video Diffusion: Consistent Multi-video Generation with Camera Control."  arXiv:2405.17414 (2024).
>
> [4] Zhang, David Junhao, et al. "ReCapture: Generative Video Camera Controls for User-Provided Videos using Masked Video Fine-Tuning." arXiv:2411.05003 (2024).
>
> [5] Xu, Dejia, et al. "Cavia: Camera-controllable Multi-view Video Diffusion with View-Integrated Attention." arXiv:2410.10774 (2024).
>
> [6] Cheong, Soon Yau, et al. "Boosting Camera Motion Control for Video Diffusion Transformers." arXiv:2410.10802 (2024).

---

> ### Author Response · Authors · 2024-11-27
>
> Dear reviewer FegY,
>
> Thank you for your time to provide the reviews for our paper. I want to remind you that I have submitted my rebuttal to your reviews,  specifically addressing the key concerns you raised regarding the novelty of CameraCtrl, the potential degeneration of object motion, and the discussion of recent camera control methods.
>
> I have provided the detailed explanations and quantitative results to clarify these points and look forward to your feedback.
>
> Thank you for your time and consideration.

---

> > ### Comment · Reviewer_FegY · 2024-11-30
> >
> > Thank you to the authors for their responses. I am OK with Question1 and 3 but the quantitative results partially addressed my concerns about Question 2.  The video results indicate that the object lacks significant subject motion. As a result, I have maintained my score.

---

> > > ### Author Response · Authors · 2024-12-01
> > >
> > > Thank you for your reply and support regarding our paper's acceptance.
> > >
> > > CameraCtrl is designed to add camera viewpoint control while working with frozen base video generation models. The degree of motion dynamics is primarily constrained by these underlying base generators, and the limitation in subject motion is a common challenge observed across concurrent models, like CamCo, VD3D, and CVD. We anticipate that future improvements in base video generators will enable more significant subject motion.
> > >
> > > We sincerely appreciate your time and constructive feedback throughout this review process.

---

### Author Response · Authors · 2024-11-23

We appreciate the thorough review and constructive feedback provided on our work. We are happy to see that the reviewers recognize the importance of our work for the video generation field (Reviewers 7FvC, BKUF, UQ23) and other related fields (7FvC, BKUF). Our work is well motivated (Reviewers FegY, UQ23). Besides, our method is simple and effective (Reviewers 7FvC, BKUF, UQ23). Additionally, the reviewers have commended our experimental and evaluation design (Reviews FegY, 7FvC, BKUF), the great performance of our model in when comparing it with other methods in different settings (Reviews 7FvC, BKUF, UQ23), and our paper is well written (Reviews FegY, BKUF).

---

### Meta-Review · Area_Chair_gVkK · 2024-12-18

**Metareview:**

(a) Claims and Findings

This paper targets the challenge of controlling camera pose in text-to-video (t2v) generation models. It introduces the use of Plücker embeddings for camera trajectory representation and proposes an adaptor-like module that can encode camera information independent of appearance. Experimental evaluations on two t2v models demonstrate the effectiveness of this approach.

(b) Strengths

The reviewers find that the paper addresses an important problem in text-to-video generation. The paper is well written, and the proposed method is both simple and effective. The extensive experiments and ablation studies provide strong empirical evidence supporting the method’s effectiveness.

(c) Weakness
* The technical novelty is limited, where the main distinction with prior works come from adopting Plücker embedding as the camera representation.
* The properties of Plücker embedding may not perfectly align with the intended camera control task
* The datasets used in the experiments are biased toward static scene and limits the dynamics in resulting videos
* A fundamental issue in the problem formulation, as pointed out by reviewer UQ23, remains unresolved

(d) Reasons for decision

On the positive side, the paper addresses an important problem, is well presented, and offers a method that is straightforward and supported by substantial empirical evidence. On the negative side, however, the paper lacks technical novelty, and the fundamental concerns about the problem formulation remain unaddressed. If one accepts the problem as currently framed (following prior works), the paper is well written and provide solid improvement for camera control in text-to-video generation model and meets the acceptance criteria. However, the AC agrees with reviewer UQ23 that a more thoughtful and principled problem formulation is strongly desired.

**Additional Comments On Reviewer Discussion:**

During the rebuttal period, the authors addressed several concerns raised by the reviewers:
* The authors provided empirical evidence supporting the superiority of Plücker embeddings over alternative representations. This resolved the reviewers' concerns regarding the representation.
* Despite limited technical novelty, reviewers acknowledged the value of the paper’s contribution to the problem of camera control in video generation.
* The authors address the concern regarding object motion by providing additional comparison based on ODD score. However, reviewers remained partially unconvinced, particularly due to limitations in the qualitative results.
* The rebuttal did not adequately address concerns regarding the problem formulation, which remains the primary unresolved issue for the paper.

---

### Decision · Program_Chairs · 2025-01-22

Accept (Poster)